

# Quantifying hydrological impacts of compacted sandy subsoils using soil water flow simulations: the importance of vegetation parameterization

Jayson Gabriel Pinza[1,2], Ona-Abeni Devos Stoffels[1], Robrecht Debbaut[1], Jan Staes[1], Jan Vanderborght[2,3], Patrick Willems[4], Sarah Garré[5]

[1]ECOSPHERE Research Group, Department of Biology, University of Antwerp, 2610 Antwerp, Belgium
[2]Division of Soil and Water Management, Department of Earth and Environmental Sciences, KU Leuven, B-3001 Leuven-Heverlee, Belgium
[3]Agrosphere Institute, IBG-3, Forschungszentrum Jülich GmbH, 52428 Jülich, Germany
[4]Urban and River Hydrology and Hydraulics Section, Department of Civil Engineering, KU Leuven, B-3001 Leuven-Heverlee, Belgium
[5]Flanders Research Institute for Agricultural, Fisheries and Food Research (ILVO), 9090 Melle, Belgium

*Correspondence to*: Jayson Gabriel Pinza (jaysonpinza@gmail.com)

**Abstract.** Numerical models can quantify subsoil compaction's hydrological impacts, useful to evaluate water management measures for climate change adaptations on compacted subsoils (e.g., augmenting groundwater recharge). Compaction also affects vegetation growth, which, however, is often parameterized using only limited field measurements or relations with other variables. Our study shows that uncertainties in vegetation parameters linked to transpiration (leaf area index [LAI]) and water uptake (root depth distribution) can significantly affect hydrological modeling outcomes. We used the HYDRUS-
1D soil water flow model to simulate the soil water balance of experimental grass plots on Belgian Campine Region's sandy soil. The compacted plot has the compact subsoil at 40–55 cm depths while the non-compacted plot underwent de-compaction. Using two year soil moisture sensor data at two depths, we calibrated and validated our models of these compacted and non-compacted plots under three different vegetation parameterizations, reflecting various canopy and root growth reactions to compaction. We then simulated the water balances under future climate scenarios.

Our experiments reveal that the compacted plots exhibited lower LAI while the non-compacted plots had deeper roots. Considering these vegetations' reactions in models, our simulations show that compaction will not always reduce deep percolation, compensated by the deep rooted non-compacted case model's higher evapotranspiration. Therefore, this affected vegetation growth can also further influence the water balance. Hence, hydrological modeling studies on (de-)compaction
should dynamically incorporate vegetation growth above- and belowground, of which field evidence is vital.



## Graphical Abstract

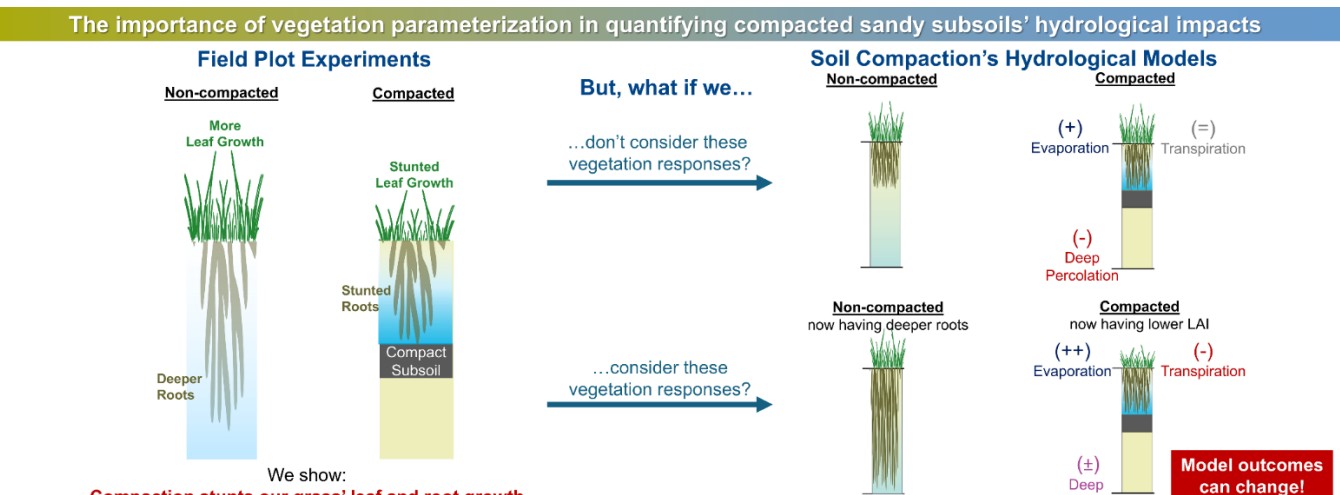



## 1 Introduction

Soil compaction has been a persistent worldwide issue that has reduced agricultural yields (Ishaq et al., 2001; Saqib et al., 2004) and even forest growth rates (Nawaz et al., 2013). It comprises at least 17% of anthropogenic soil degradation cases in Europe and 4% worldwide (Alakukku, 2012; Oldeman et al., 1991). It occurs when soils are subjected to stresses exceeding their strength (Dexter, 1988). The sources of soil stresses could be natural (e.g., drying/freezing, rainfall, roots, soil mineralogy and parent material, higher clay content present, foot traffic, animal grazing) or artificial (e.g., machinery) (Houšková and Montanarella, 2008; Nawaz et al., 2013; Shaheb et al., 2021; Yang et al., 2022; Zhao et al., 2007). These stresses lead to increased soil bulk densities and reduced porosities and infiltration rates (Nawaz et al., 2013; Silva et al., 2008; Soil Science Society of America, 2008). In turn, these changes in physical parameters can stunt plant growth in terms of height, biomass, roots (depth, length, penetration), and leaf growth (leaf area) (Gliński and Lipiec, 2018; Kristoffersen and Riley, 2005; Nawaz et al., 2013; Passioura, 2002; Shah et al., 2017; Shaheb et al., 2021). Compaction is more persistent in subsoils than topsoils because the natural alleviating processes (wetting/drying, freezing/thawing, root growth) rapidly diminish with depth (van den Akker and Schjønning, 2003; Batey, 2009).

Compaction also affects soil hydrological processes. Topsoil compaction promotes surface runoff especially during heavy precipitation events because of hindered vertical infiltration by reduced macropore volumes (Alaoui et al., 2018; Byrd et al., 2002). Moreover, it also promotes evaporation due to the small pores that favor more capillary flow (Goldberg-Yehuda et al., 2022; Romero-Ruiz et al., 2022). For subsoils, compact layers (such as plow or tillage pans) have a higher bulk density and a lower total porosity (smaller and more isolated pores) than the soil directly above or below it (Bertolino et al., 2010; Gliński et al., 2011). This then prevents surplus water from further percolating (Adekalu et al., 2006; Allmaras, 2003; Bertolino et al., 2010), thereby increasing soil moisture on the zones above the dense layer (Moreno et al., 2003; Nawaz et al., 2013) and lateral flow above the plow pan (interflow) and runoff (Alaoui et al., 2018; Jiang et al., 2015). With more retained soil moisture, more water is available for evaporation (Agrawal, 1991; Assaeed et al., 1990; Hoefer and Hartge, 2010) and root water uptake (and thus transpiration). However, the stunted root growth brought by compaction reduces soil water uptake in the dense subsoil and deeper zones (Bengough et al., 2011; Passioura, 2002; Wang et al., 2019). This is then compensated by increased uptake in looser soils overlying the dense subsoil, attributed mainly to increased soil-root contact and higher unsaturated hydraulic conductivity promoting more water flow (Andersen et al., 2013; Lipiec and Hatano, 2003; Nosalewicz and Lipiec, 2014). Moreover, stunted leaf growth, manifested by reduced leaf areas, hindered transpiration (Assaeed et al., 1990; Grzesiak, 2009; Umaru et al., 2021). Subsoil compaction also hinders deeper percolation and hence groundwater recharge (Negev et al., 2020; Owuor et al., 2016; Radatz et al., 2012). De-compaction is thus recommended to promote recharge (Garcia and Galang, 2021; Priori et al., 2020; Tarigan et al., 2020). In short, subsoil compaction leads to higher accumulated soil moisture above the compact subsoil layer, higher runoff, and lower deep percolation, but transpiration and evaporation can be higher or lower with compaction (Fig. 1, Table 1).



These conclusions from experimental observations were further confirmed by hydrological modeling studies that quantify compaction's impacts on soil water fluxes. These models are also useful to simulate hydrological impacts that can guide water resource management under climate change scenarios. However, accurate model simulations require accurate estimates of model parameters that should be based on accurate measurements (Moreno et al., 2003). Given that transpiration has a large impact on the soil water, vegetation parameters related to root water uptake and leaf area are thus important. Therefore, the impact of soil compaction on vegetation parameters should also be included in model simulations to assess the overall effect on soil water fluxes. However, vegetation parameter values were mostly based on limited field measurements (i.e., taken only at the end of harvesting period), assumptions based on correlations or derivations from other variables, or calibrations/parameter estimations. Some of these parameters were even assumed to be the same for compacted and non-compacted setups (Table 1), examples of which are leaf area indices (LAI) (Moreno et al., 2003; Voter and Loheide, 2018) and root development (Hartmann et al., 2012; Romero-Ruiz et al., 2022).

These modeling studies also focused mostly on fine soil textures (i.e., silty, loamy). Only one hydrological modeling study on compaction involved sandy soils (Moreno et al., 2003) (Table 1), which reported negligible hydrological impacts of compaction. But, sandy subsoils exhibit higher susceptibility to machinery-induced compaction than loam or clay subsoils (Jones et al., 2003; Scanlan et al., 2022; Spoor et al., 2003). Moreover, sandy soils are globally relevant, occupying 31% of total global area comprising all continents (De Holanda et al., 2023; Huang and Hartemink, 2020).

Hence, for these less studied sandy soils, we show how such uncertainties in vegetation parameters related to transpiration and root water uptake can largely affect the modeling outcomes. Here, we performed pilot experiments involving vegetated plots with intact compact sandy subsoil and plots which underwent de-compaction. We then used calibrated and validated 1D soil water flow models (HYDRUS), parameterized based on these experiments, to disentangle these plots' water budgets under various climate scenarios.



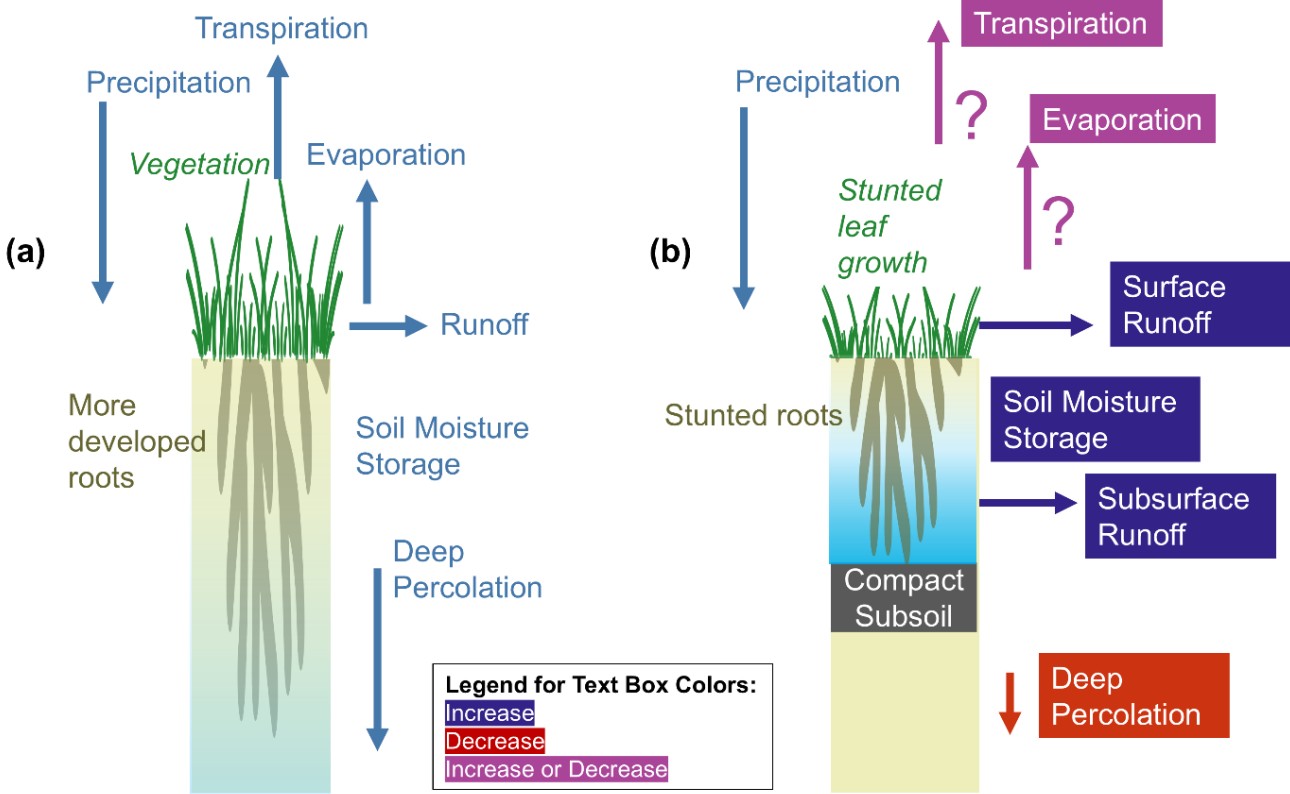

**Figure 1. Summary of compaction's known impacts on soil water hydrology, comparing (a) non-compacted case and (b) compacted case with compact subsoil.**



**Table 1. Numerical hydrological modeling studies on soil (de-)compaction's hydrological impacts in small-scale sites**

| Soil Type and Compaction Depths | Vegetation | Study Area's Size | Location | Soil Water Flow Model | Vegetation Parameters Considered | Outcome of Compaction's Impact to Hydrology and Vegetation's Productivity and Difference i.e., compacted minus non-compacted | Cited Study |
|---|---|---|---|---|---|---|---|
| Sandy soil (Cambisols) with subsoil compaction at 35–45 cm | Maize | 2 subplots of 450 m² each | Seville Province, (SW Spain) | SIMWASER | • Temporally varying root depths (measured and simulated in the models separately for compacted and non-compacted cases) <br> • Only one temporally constant LAI value applied for both compacted and non-compacted cases | Annual scale: <br><br> No impact on *ET*, *DP* and grain yield. | Moreno et al. (2003) |
| Silty loam soil (Chromic Cambisol) with topsoil compaction at 0–15 cm | No mention | No data | Heyang, Loess Plateau (SE China) | Watermod | No mention | Annual scale [mm]: <br><br> Increase: *R* (+2 to +218), *T* (+1 to +10) <br> Decrease: *DP* (-1 to -197), *E* (-1 to -39), *ET* (-41 to -43) | Epée Missé (2015) |
| Silty loam soil (unknown WRB* classification) with urban topsoil compaction until 7.5 cm | Turfgrass | 825 m² residential parcel | Madison, Wisconsin (United States) | 1D ParFlow Common Land Model (CLM) | • Constant LAI (maximum and minimum only) from literature (same for compacted and non-compacted cases) <br> • Calibrated constant root fitting parameters based on modified values from literature (same for compacted and non-compacted cases) | Growing season scale (April–November) [mm] <br> Increase: *R* (0 to +126.3), *ET* during an average (non-dry) year (+0.6 to 4.0) <br> Decrease: *DP* (-1.6 to -128.6), *ET* during a dry year (-9.2 to -50.0) <br><br> This study is executed by de-compacting already compact soil rather than inducing compaction in the field. Moreover, de-compacting could have been performed in conjunction with other measures (e.g., disconnection with impervious surfaces, etc.) | Voter & Loheide (2018) |





| Soil Type and Compaction Depths | Vegetation | Study Area's Size | Location | Soil Water Flow Model | Vegetation Parameters Considered | Outcome of Compaction's Impact to Hydrology and Vegetation's Productivity and Difference i.e., compacted minus non-compacted | Cited Study |
|---|---|---|---|---|---|---|---|
| Loamy silt soil (Eutric Cambisol) with topsoil and subsoil compaction at 0–30 cm | Maize | 7 m wide strip compacted by a tractor | Wieselburg (Austria) | SIMWASER | • Temporally varying root length density across the 120 cm profile (accounting for root growth) <br> • Same root length density and root depth for compacted and non-compacted case (measured in the field at harvest season) <br> • Constant potential rooting depth value from literature (same for compacted and non-compacted cases) <br> • Constant index for root "strength" class (measured in the field separately for compacted and non-compacted cases) <br> • Constant LAIs (measured in the field separately for compacted and non-compacted cases at the end of harvest season) <br> • Constant extinction coefficient of visible radiation (measured in the field separately for compacted and non-compacted cases) <br> • Calibrated constant stomatal resistance | 6 month scale [mm for hydrological components; kg ha$^{-1}$ for biomass]: <br><br> Increase: $R$ (+37 to +90 mm), $SMS$ (+2 to +30 mm) <br><br> Decrease: $DP$ (-12 to -16), $ET$ (-26 to -105), dry biomass (+62 to -4884 kg ha$^{-1}$) | Stenitzer & Murer (2003) |
| Silty soil (Stagnic Luvisol) with topsoil and subsoil compaction until 30 cm depth(?) | 5 years: Winter wheat + Annual winter barley (2 years), winter rye, sugar beet | No data | North Rhine-Westphalia (Germany) | HYDRUS-1D | • Temporally varying (growing) crops and roots from April to July and August to November (same parameter values for compacted and non-compacted cases) <br> • Impacts of compaction on root development were not considered | Growing season scale (April–July) [mm] <br> Increase: $T$ (+49 to +60) [1991–2000; 2051–2060; 2091–2100] <br> Almost no impact: $E$ (maximum 3 mm difference for 3 scenarios) <br><br> August–March [mm] <br> Almost no impact: $T$, $E$ | Hartmann et al. (2012) |





| Soil Type and Compaction Depths | Vegetation | Study Area's Size | Location | Soil Water Flow Model | Vegetation Parameters Considered | Outcome of Compaction's Impact to Hydrology and Vegetation's Productivity and Difference i.e., compacted minus non-compacted | Cited Study |
|---|---|---|---|---|---|---|---|
| Silty loam (Cambisol) with topsoil compaction until 30 cm depth | Treatments with no vegetation and others with grass-legume mixtures | 170–204 m² plot | Zurich (Switzerland) | HYDRUS-1D | Root distribution functions are depth dependent and are the same for both compacted and non-compacted cases | Annual scale [mm]<br>Increase: $E$ (+5 to +15)<br>Decrease: $DP$ (-5), $T$ (-10)<br><br>Note: we visually estimated these values from stacked bar charts in their study. | Keller et al., (2017); Romero-Ruiz et al., (2022) |
| Silty loam soil (unknown WRB classification) with topsoil and subsoil compaction until 40 cm depth | Wheat | No data | Loess Plateau (SE China) | CoupModel | ● Plant growth is considered based on potential yield, proportional to the global radiation intercepted by canopy, and factors that hinder growth (e.g., unfavorable air temperature, nitrogen availability [assumed sufficient in this study], water stress) (same parameter values for compacted and non-compacted cases)<br>● Pests and nutrient-deficiency factors not included | Annual scale [mm]<br>Increase: $E$ (+12 to +24)<br>Decrease: $DP$ (-2 to -20), $T$ (-10 to -15) | Zhang et al. (2007) |
| Silty clay loam soil (unknown WRB classification) with subsoil compaction at 40–70 cm depth | Winter wheat (early October to mid June)<br><br>Summer maize (mid June to end September) | 990 m² | Xiongxian area (North China) | 1D WAVE model | ● LAI, crop coefficient, crop phenology based on experimental site data (measured separately for compacted and non-compacted cases)<br>● Root water uptake functions derived separately for compacted and non-compacted cases from field observations | Annual scale [mm]<br>Increase: $SMS$ (+6.1 [normal year] to +31.6 [wet year]), $E$ (+5.3 [wet year] to +12.4 [dry year])<br>Decrease: $DP$ (-5.3 [normal year] to -33.2 [dry year]), T (-0.3 [dry year] to -12.7 [wet year])<br><br>This study is executed by de-compacting (by subsoiling tillage) already compact soil rather than inducing compaction in the field. | Xu & Mermoud (2003) |

**Abbreviations: WRB=World Reference Base; LAI= Leaf Index; *ET*=Actual Evapotranspiration; *E*=Actual Evaporation; *T*=Actual Transpiration; *DP*=Deep Percolation; *SMS*=Soil Moisture Storage; *R*=Runoff**



## 2 Materials and Methods

### 2.1 Experimental Site and Setup

Our site is located southeast of Lille, Province of Antwerp, northern Belgium (Appendix A – Fig. A1) within the Kleine Nete watershed of the Campine Region. The site elevation is about 10 m above sea level. The land use near our site is grassland and arable land (maize and potatoes are the main crops) and the site has been under grasslands since 1970s. The site exhibits temperate climate. Over years 1979–2023, annual precipitation and potential evapotranspiration values range from 480–890 and 500–800 mm respectively (Toreti, 2014) (Fig. 2a, Fig. 2b).


The site's soil is loamy sand with anthropogenic humus A horizon (Flemish Environmental Planning Agency, 2005). A compacted subsoil at 40 to 55–60 cm depth is present, which is an iron oxide pan (Bsm horizon) having green color and red-orange concretions (Appendix A – Fig. A2). This pan is also reported to be common on sandy soils found east of Antwerp, Belgium (Vandamme, 1978), where our study site is situated. Other soil types within the site's vicinity are in Appendix A –

Fig. A3.

Eight 2 m square plots were constructed at the site in December 2021. Four of them still have the compact layer intact at 40 cm downwards. The remaining were de-compacted by excavating the soil down to 150 cm depth using a crane to eliminate the compact layer (Appendix A – Fig. A4, A5). For each plot, the excavated sandy topsoil layer (40 cm thick for compacted

plots and 150 cm thick for non-compacted plots (Fig. 3)) was loosened and homogenized well using a cement mixer and then returned to that same plot. In four plots, 480 L woodchips were also mixed, mimicking Campine Region's general agricultural practice (Appendix A – Fig. A4). Penetrometer logs were also used to check the effectiveness of de-compaction wherein the lack of peaks in pressure values indicate the absence of compacted layer (Fig. 3a). Impermeable plastic boxes were installed to avoid subsurface lateral in or outflow (Appendix A – Fig. A5). Per plot, two capacitance sensors (HOBO

Onset S-SMC-M005) monitored the soil moisture content at 10 and 40 cm depths, allowing to capture the hydrological conditions near the surface and right above the compact subsoil layer, respectively (Fig. 3). This monitoring spanned from 08 July 2022 to 08 September 2024.

Each plot had either maize *(Zea mays)* or mixtures of Italian ryegrass *(Lolium multiflorum)* and white clover *(Trifolium*

*repens)* as vegetation cover (Appendix A – Fig. A4), thus replicating Campine Region's general agricultural landscape. No plots were fertilized.

The grass-clover mixtures were mowed to 15 cm stubble height (mowing timeline in Fig. 2e and Appendix A – Fig. A6) whenever they appeared to grow beyond the plot boxes (Fig. 3). As shown in Eq. (1), the clippings' dry weights were





converted into *LAI* using a representative leaf dry mass of 66 g m$^{-2}$, based on the mean of the median leaf dry mass of herbs (60 g m$^{-2}$, representing white clover) and of graminoids (72 g m$^{-2}$, representing grass) (Poorter et al., 2009)

$$LAI = \frac{\left(\frac{dried\ weights\ of\ grass-clover\ mixture}{mean\ leaf\ dry\ mass\ per\ unit\ leaf\ area}\right)}{plot\ area} \tag{1}$$

Temporal hourly *LAI* values during grass-clover's growths (Fig. 2e) were linearly interpolated from the *LAI* after last mowing and *LAI* before the next mowing. The *LAI* after last mowing, pertaining to the grass remaining after mowing, was always assumed to be 0.25, significantly lower than the lowest *LAI* from clippings (0.84–1.01) (Fig. 2e). This *LAI* after the last mowing was then added with *LAI* from the next mowing's weighed clippings to obtain the overall *LAI* before last mowing.


Maize was sown on 26 April 2022; 11 May 2023; and 22 May 2024 and then harvested on the onset of wet season on 17 October 2022; 20 October 2023; and 08 October 2024. Harvesting was done by cutting them down, leaving 10 cm stem above ground level, whereafter the maize plots were left fallow.

Outside the plots, we also obtained eight non-compacted sand and seven compact sand samples at 15- and 50 cm depths, respectively. Their grain size compositions were estimated by laser diffraction technique. Their bulk densities were estimated based on their dry masses and ring kit volumes. Saturated hydraulic conductivity *(K$_s$)* was estimated from the falling head method (Table 2).

**Table 2. Laboratory measurements of soil properties. Values in parenthesis denote the geometric mean for *K$_s$* and arithmetic mean for the rest.**

| Layer [sampling depth] | Grain size composition [%] | Dominant grain size classification* and diameter [μm] | Bulk density [g cm$^{-3}$] | $K_s$ [cm d$^{-1}$] |
|---|---|---|---|---|
| Loose sand [15 cm depth] | Sand: 83.7–85.4 (84.5) Silt: 11.9–13.3 (12.7) Clay: 2.5–3.1 (2.8) | fine sand 201–222 (208) | 1.22–1.44 (1.37) | 26.71–977 (202.56) |
| Compact sand [50 cm depth] | Sand: 85.0–86.1 (85.5) Silt: 11.5–12.6 (12.0) Clay: 2.4–2.5 (2.5) | fine sand 201–215 (204) | 1.48–1.65 (1.59) | 0.634–111 (7.35) |

*Based on Harmonized world soil database version 2.0 of FAO (Nachtergaele et al., 2023)

Precipitation, air temperature (Fig. 2c), solar radiation, relative humidity, and average wind velocity were measured on an
hourly resolution using our weather station located 2 km from the site. Measurement gaps were filled using the data from the closest active meteorological station at Herentals (MOW-HIC et al., 2021), 5.5 km away from the site. Daily mean temperature ranged from -5.2 °C (in December 2022) to 27.5 °C (in July 2022) during our experiments (Fig. 2c). From these



meteorological variables, potential evapotranspiration ($ET_0$) [mm h$^{-1}$] was derived using the Food and Agriculture Organization (FAO) 56 Penman-Monteith Equation for a reference crop (hypothetical grass surface) (Allen et al., 1998), shown in Eq. (2).

$$ET_0 = \frac{0.408\Delta(R_n - G) + \gamma U_2(e_s - e_a)(\frac{37}{T+273})}{\Delta + \gamma(1 + 0.34U_2)}$$

$$G = 0.1R_n \,, R_n > 0 \ \ [daytime]$$ 

$$G = 0.5R_n \,, R_n \leq 0 \ [nighttime]$$

(2)

where $\Delta$ is the slope of the vapor pressure curve [kPa °C$^{-1}$] , $R_n$ is the net radiation at the grass' surface [(MJ m$^{-2}$ h$^{-1}$], $G$ is the soil heat flux [(MJ m$^{-2}$ h$^{-1}$] whose calculation depends on the sign of $R_n$ that indicates if the hourly $R_n$ was measured at daytime or nighttime, $\gamma$ is the psychrometric constant [kPa °C$^{-1}$], $U_2$ is the wind speed measured at 2 m height [cm s$^{-1}$], $e_s$-$e_a$ is the vapor pressure deficit [kPa] and $T$ is the average air temperature [°C]. The resulting daily potential evapotranspiration and comparison with precipitation are shown in Fig. 2c and Fig. 2d.

In 2022, we mimicked intense summer rainfall events that could generate surface runoff and could be more frequent in the future climate (Hodnebrog et al., 2019; Hosseinzadehtalaei et al., 2020) by irrigating the plots. Applied irrigation doses are listed and justified in Appendix B – Table B1 and Table B2.

At the end of the monitoring period, we then sampled the roots inside plots with grass-clover mixtures. For each of these plots, a root length density profile and a dry biomass profile were obtained. Root-soil cores (8 cm diameter, 15 cm height) were extracted every 15 cm throughout the 0–105 cm depths using root augers. These cores were then washed thoroughly to obtain the roots. The root lengths were estimated using the grid-intersection method and Eq. (3) (Freschet et al., 2021; Newman, 1966; Tennant, 1975). These roots were also oven-dried at 70 °C for a week and then weighed.

$$Root\ length = \frac{11}{14} \times total\ number\ of\ intercepts\ (horizontal\ and\ vertical)\ \times grid\ unit\ length$$

(3)

The timeline of our experiments is summarized in Appendix A – Fig. A6.



**Figure 2. Monthly boxplots of (a) precipitation and (b) potential evapotranspiration within years 1979–2023 at the 25-km square grid centered at 51.16065° N, 4.87048° E (Olen, Belgium) (Toreti, 2014). (c) Daily potential evapotranspiration, precipitation, mean temperatures, and applied irrigation amounts during the experiments, (d) Daily precipitation deficits, (e) Daily leaf area index (*LAI*) for grass-clover mixtures of non-compacted and compacted cases and mowing timeline.**





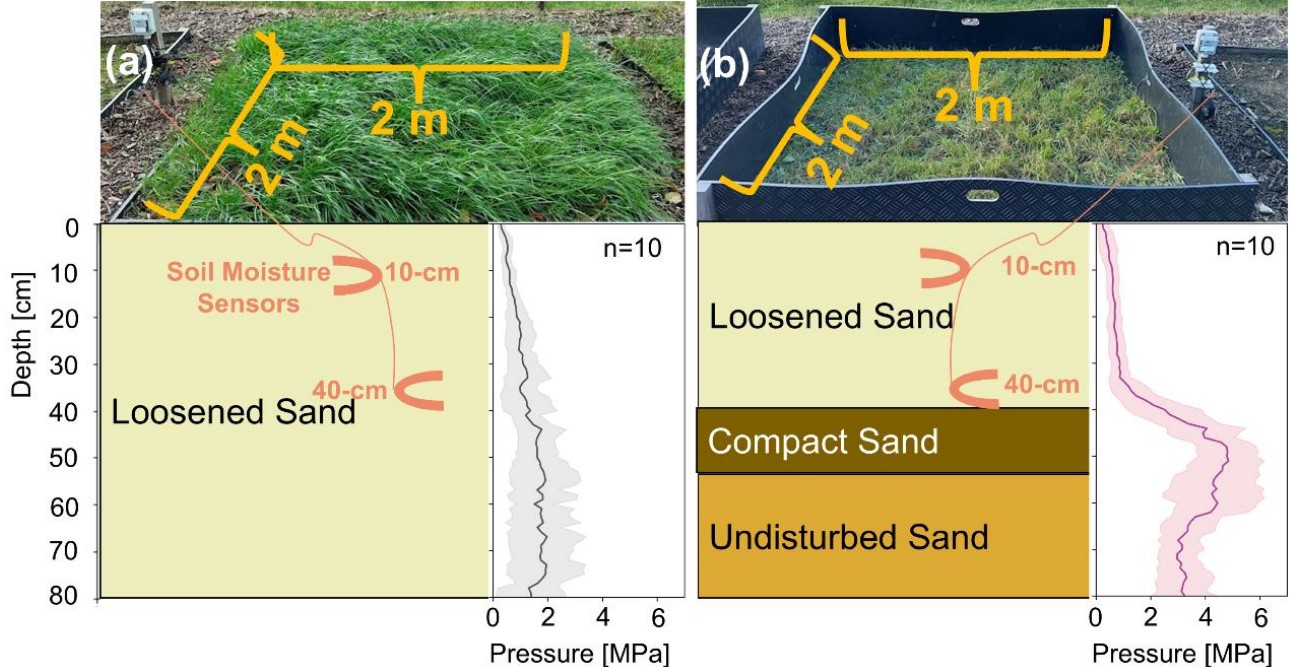

**Figure 3. Experimental plots, schematic soil profiles, and averaged penetrologger profiles (± standard deviation, represented by shaded bands) for (a) a non-compacted case and (b) a compacted case.**

## 2.2 Numerical Model Setup

We represented the experimental plots as 1D soil columns, assuming that lateral flow there was insignificant (Fig. 3). The soil water hydrology was modeled using the numerical model HYDRUS-1D (Šimůnek et al., 2008).

### 2.2.1 Soil Water Movement

The HYDRUS-1D code simulates 1D (vertical) water movement in porous media by numerically solving the Richards equation for water flow along variable-saturated media, neglecting air phase and thermal gradient-driven flow (Richards, 1931; Šimůnek et al., 2008), shown in Eq. (4).

$$\frac{\partial \theta}{\partial t} = \frac{\partial}{\partial z}\left[K(h)\left(\frac{\partial h}{\partial z} + 1\right)\right] - S(h) \tag{4}$$

where $h$ is the water pressure head [L], $\theta$ is the soil moisture content (SMC) [L$^3$ L$^{-3}$], $t$ is time [T], $z$ is the spatial coordinate [L], $S(h)$ is the sink term [L$^3$ L$^{-3}$ T$^{-1}$] that accounts for root water uptake and $K(h)$ is the unsaturated hydraulic conductivity function [LT$^{-1}$].



The van Genuchten-Mualem functions were used to describe the $\theta$(h) and $K$(h) relations (Mualem, 1976; Van Genuchten, 1980), shown in Eq. (5):

$$\theta(h) = \begin{cases} \theta_r + \dfrac{\theta_s - \theta_r}{[1 + |\alpha h|^n]^m} & h < 0 \\ \theta_s & h \geq 0 \end{cases}$$

$$K(h) = K_s S_e^L \left[ 1 - \left(1 - S_e^{-1/m}\right)^m \right]^2 \tag{5}$$

$$S_e = \frac{\theta - \theta_r}{\theta_s - \theta_r}$$

$$m = 1 - \frac{1}{n}, n > 1$$

where the $\theta_s$ and $\theta_r$ are the saturated and residual soil water content, respectively. $K_s$ is the saturated hydraulic conductivity [L T$^{-1}$]. $S_e$ is the effective water content. $\alpha$ [L$^{-1}$] and $n$ [-] are empirical parameters of the water retention curve functions. Together with the tortuosity and pore connectivity parameter $L$, these empirical parameters influence the hydraulic functions' shape. In our models, $L$ was set to 0.5, the value frequently applied for mineral soils (Dettmann et al., 2014; Mualem, 1976). The Richards equation is then solved numerically at each node using Galerkin-type linear finite element schemes.

For the root water uptake, the water volume removed from a unit soil volume per unit time (sink term, $S$) is defined using Eq. (6) (Feddes et al., 1978; Šimůnek et al., 2008)

$$S(h) = \alpha(h)S_P \tag{6}$$

where the $\alpha$(h) [-] is a prescribed function of the soil water pressure head ($0 \leq \alpha \leq 1$) and $S_P$ [T$^{-1}$] is the potential water uptake. $S_P$ is further defined using Eq. (7):

$$S_P = T_P b(z) \tag{7}$$

where $T_P$ is the potential transpiration rate [LT$^{-1}$], and $b(z)$ is the normalized water uptake distribution [L$^{-1}$], which reflects $S_P$'s spatial variation throughout the root zone. The $b(z)$ is based on normalized root distributions ($b'(z)$) as shown in Eq. (8) (Šimůnek et al., 2008):

$$b(z) = \frac{b'(z)}{\int_{L_R} b'(z)dz} \tag{8}$$

$$\int_{L_R} b(z)dz = 1$$

where $L_R$ is the root zone region or domain. Solving for $T_P$ by combining Eq. (7) and Eq. (8), resulting in Eq. (9):

$$T_P = \int_{L_R} S_P dz \tag{9}$$

Meanwhile, the actual root water uptake can involve water uptake compensation along root zones. The presence of this compensation depends on the dimensionless water stress index $\omega$, defined in Eq. (10), and a user-defined threshold $\omega_c$ (also known as the critical water stress index or root adaptability factor) (Jarvis, 1989; Šimůnek et al., 2008).

$$\omega = \int_{L_R} \alpha(h, z)b(z)dz \tag{10}$$





When $\omega$ exceeds $\omega_c$, reduced root water uptakes in stressed zones within $L_R$ are compensated by other zones' increased uptakes. Otherwise, no compensation occurs. Following Eq. (11), $\omega_c < 1$ promotes compensation (fully at $\omega_c = 0$) while $\omega_c = 1$ inactivates it (Šimůnek et al., 2008).

$$S(h, z) = \begin{cases} \alpha(h, z) b(z) \frac{T_P}{\omega_c}, & \omega < \omega_c \\ \alpha(h, z) b(z) \frac{T_P}{\omega}, & \omega_c < \omega < 1 \end{cases} \tag{11}$$

$$\frac{T_a}{T_P} = \begin{cases} \frac{\int_{L_R} \alpha(h,z) b(z) \, dz}{\omega_c} = \frac{\omega}{\omega_c} < 1, & \omega < \omega_c \\ \frac{\int_{L_R} \alpha(h,z) b(z) \, dz}{\omega} = \frac{\omega}{\omega} = 1, & \omega_c < \omega < 1 \end{cases}$$

### 2.2.2 Boundary Conditions

The upper boundary condition (atmospheric boundary condition with surface runoff) (Fig. 4) involved meteorological forcings related to precipitation and potential evapotranspiration. For precipitation, the hourly values recorded by the site's weather station were used. For potential evapotranspiration, the hourly values *(ET₀)* were calculated using Eq. (2) and then partitioned to potential transpiration ($T_P$) and potential soil evaporation ($E_P$) in HYDRUS based on LAI in accordance to the Beer's Law, shown in Eq. (12) (Ritchie, 1972; Šimůnek et al., 2008).

$$T_P = ET_0 \times \left(1 - e^{-rExtinct(LAI)}\right)$$
$$E_P = ET_0 - T_P = ET_0 \times e^{-rExtinct(LAI)} \tag{12}$$

where *rExtinct* is the radiation extinction coefficient, equal to 0.463 (Šimůnek et al., 2008). The LAI time series for non-compacted or compacted plots (Fig. 2e) were used to parameterize the corresponding case's $ET_0$ partitioning.

Runoff was calculated whenever the pressure head reached zero at the soil surface. Ponding was disregarded to simplify the quantification of this excess water in our simulations.

The lower boundary condition was set to free drainage (zero gradient boundary condition (Šimůnek et al., 2008)) (Fig. 4). This is based on the water table being significantly deeper (200 cm) than the compact layer's bottom depth (55–60 cm) as observed from our soil profiles from March 2023 and March 2024. Even when considering capillary rise (+13.5 to +50 cm for fine sands (Fetter, 1994)), the capillary fringe was likely still deeper than the compact layer.

### 2.2.3 Vegetation Parameterizations

We considered grass as the models' vegetation. Since the grass had been present throughout the experimental period, root distributions are assumed to be constant in time. Root water uptake compensation (Jarvis, 1989; Šimůnek et al., 2008) was



also permitted using $\omega_c$ = 0.1. This is because the default Feddes model parameter values for grasses in HYDRUS-1D (Feddes et al., 1978; Šimůnek et al., 2008) lead to underestimated root water uptake values for coarse-textured soils (Peters et al., 2017) like the sandy soils in our study. These underestimations are relevant especially when simulating dry and wet water stress conditions.

In setting up the model, three approaches on vegetation parameterizations were considered to examine the impact of varying vegetation parameters (likely affected by compaction) on the soil water balance components. For each vegetation parameterization, two setups (non-compacted and compacted) were represented with the compact layer at 40–55 cm depths. For each model setup, the soil profile ranged from 0–100 cm (Fig. 4), having 211 nodes. The 0–55 cm depth had constant fine spatial resolution of ⅓ cm because this depth involved more relevant hydrologic processes (e.g., soil evaporation,
runoff), the observation points at 10 and 40 cm, and an additional compact layer for the compacted case models. The remaining 55–100 cm depths had constant coarser 1 cm resolution to save computational time and power.
In Vegetation Parameterization 1, we assumed that compaction does not affect the vegetation, i.e., no effect on LAI and root depth. Since we observed that no roots develop in the compact layer (Appendix A – Fig. A7), the roots were limited to 40 cm depths in both compacted and non-compacted case models (Fig. 4a, Fig. 4b). This is also the typical root biomass
distribution of Italian ryegrass in loamy clay Cambisol under a temperate oceanic climate (Durand et al., 2010; Kunrath et al., 2015) and in a loamy soil under phytotron conditions (24 °C day, 22 °C night, relative humidity 80%, 16 h photoperiod) (Lambrechts et al., 2014). The non-compacted plots' LAI time series (Fig. 2e) was adopted for both the compacted and non-compacted case models. With this, transpiration has more weight than soil evaporation even in the compacted case model.

In Vegetation Parameterization 2, we assumed again the same root distribution in the compacted and non-compacted case models. However, we also considered the effect of the soil compaction on the LAI (Fig. 4c, Fig. 4d) and thus used the respectively measured LAI time series for each case (Fig. 2e). Compared to the previous parameterization, less weight is given to transpiration than evaporation in the compacted case model. Furthermore, this model represents yield decrease in compacted soils, also reported for sandy subsoils (Laker, 2001; Laker and Nortjé, 2020; Pumphrey, 1980).

In Vegetation Parameterization 3, both the LAI and root distribution varied between the compacted and non-compacted case models (Fig. 4e, 4f). We considered the same setup as Vegetation Parameterization 2 for the compacted case model. However, we applied a deeper (i.e., 100 cm depth) root distribution for the non-compacted case model (Fig. 4e). These are based on observed larger root length and biomass densities and deeper root depths for the non-compacted plots (Appendix A
– Fig. A7). This also represents the vegetation's response to the lower water availability in sandy soils by developing a deeper root zone with more access to deeper soil water especially under drier periods. Meanwhile, the compacted setup's roots cannot penetrate in the subsoil (Harrison et al., 1994; Vanderhasselt et al., 2024). This vegetation parameterization





allows to reduce the drought stress and leads to more transpiration in the non-compacted soil than in the previous vegetation parameterizations.


### 2.2.4 Soil Hydraulic Properties: Model Calibration and Validation

For model calibration, we selected a single soil hydraulic parameter set for all vegetation parameterizations based on HYDRUS-1D's inverse modeling of both non-compacted and compacted cases under Vegetation Parameterization 3. With this, no differences in soil parameters can contribute to the differences between these vegetation parameterizations' results.

The period of 08 July 2022 (the start of soil moisture monitoring) until 30 September 2023 was selected for calibration, covering both wet and dry periods. The next hydrological year (01 October 2023 to 08 September 2024) was used for model validation.

The constraints of residual soil water content ($\theta_r$), saturated soil water content ($\theta_s$), $\alpha$ and $n$ (Appendix B – Table B3) were

derived from the measured grain size distribution and bulk densities using ROSETTA pedotransfer function (version 2) (Schaap et al., 2004). The constraints for $K_s$ of loose and compact sand layers were based on falling head permeameter tests (Table 2). The hydraulic properties of the loose sand layers in both non-compacted and compacted plots were assumedly the same, given their similar grain size composition (Table 2) and complete homogenization in the field.

The soil moisture time-series used for inverse modeling are averages of observations taken across compacted and de-compacted plots (Appendix A – Fig. A8), regardless of the applied treatments (Appendix A – Fig. A4). The initial soil moisture content values throughout the profiles were based on linear interpolation involving these averaged soil moisture content measurements at 10 and 40 cm depths (corresponding to last hour before the first hour of simulations). Mean soil moisture measurements during hours with below 0 °C air temperatures were discarded as they might not reflect the actual

moisture content under freezing conditions.

The loose soil's parameter values were derived first using observations from the non-compacted plots. These obtained values were then adopted and fixed for the compacted case's loose layers so that only the compact subsoil layer's parameters remained to be optimized via inverse modeling using the compacted plots' mean soil moisture time series. The resulting

water retention and conductivity curves depicting the estimates from the inverse models and from the ROSETTA functions for field samples are in Appendix A - Fig. A9, Fig. A10.

Table 3 summarizes the parameter values and settings used in our non-compacted and compacted case models under all vegetation parameterizations.






**Figure 4. Schematic diagram of 1D soil water flow models for non-compacted and compacted cases under the three vegetation parameterizations.**

[1]LAI adopted only from the non-compacted case in the experimental plots (Fig. 2e)

[2]Rooting depth reaches to 40 cm and distribution is 1 from 0–10 cm depths and linearly decreasing to 0 from 10 to 40 cm depths

[3]LAI adopted from non-compacted and compacted cases in the experimental plots, respectively (Fig. 2e)

[4]Non-compacted case model: distribution is 1 (0–10 cm depths), then linearly decreasing to 0 (10 to 100 cm depths); compacted case: distribution is 1 from 0–10 cm depths and linearly decreasing to 0 from 10 to 40 cm depths




**Table 3. Model parameter specifics and justifications applied to all vegetation parameterizations**

| | | |
|---|---|---|
| **Simulations** | | |
| Process | ● Soil Water Flow<br>● Root Water Uptake<br>● No Root Growth | To simulate soil water flow with transpiration under constant rooting depths as the grasses have been present since the start of the experimental period (Appendix A – Fig. A6) |
| **Model Domain** | | |
| Profile Depth | 100 cm | To adopt the rooting depth of Vegetation parameterization 3's non-compacted case model that shows deeper root distribution |
| Spatial Resolution | ● ⅓ cm at 0–55 cm depths<br>● 1 cm at 55–100 cm depths<br>● Number of nodes = 211 | Finer resolution for shallower zones (0–55 cm) depths where observation points and compact layer are present |
| Temporal Resolution | ● Hourly for model parameterization<br>● Daily for scenario analyses | Based on temporal resolution of recorded precipitation and calculated potential evapotranspiration from<br>-our site's weather station (for model parameterization), and<br>-from historical and future climate projection data (for scenario analyses) |
| **Boundary Conditions** | | |
| Upper | Atmospheric (with Runoff) | Allows to input precipitation and potential evapotranspiration from<br>-weather station (hourly) (Fig. 3), and<br>-historical and future climate projection data (daily) in scenario analyses.<br>Runoff immediately occurs beyond saturation of topmost soil profile |
| Lower | Free Drainage | Water table and capillary fringe were observed to be deep from soil profiles (200 cm in March 2023 and March 2024) |
| **Vegetation Parameters** | | |
| *rExtinct* | 0.463 | Default value in HYDRUS-1D (Šimůnek et al., 2008) |
| $\omega_c$ | 0.1 | To promote root water uptake compensations across the profile based on Eq. (11), given reported underestimated transpiration in coarse soils (Peters et al., 2017) |
| P0<br>P0pt<br>P2H<br>P2L<br>P3 | -10 cm<br>-25 cm<br>-300 cm<br>-1000 cm<br>-8000 cm | Default value for grass in HYDRUS-1D (Šimůnek et al., 2008) |
| r2H<br>r2L | 0.5 cm d$^{-1}$<br>0.1 cm d$^{-1}$ | Default value in HYDRUS-1D (Šimůnek et al., 2008) |
| **Loosened Sand's Hydraulic Parameters** | | |
| $\theta_r$<br>$\theta_s$<br>$\alpha$<br>N | 0.0427 cm$^3$ cm$^{-3}$<br>0.407 cm$^3$ cm$^{-3}$<br>0.0544 cm$^{-1}$<br>1.457 | Optimized using inverse modeling under vegetation parameterization 3; constraints derived from measured grain size compositions via ROSETTA 2 |
| $K_s$ | 297.672 cm d$^{-1}$ | Optimized using inverse modeling; constraints from head test measurements |
| L | 0.5 | Usual value for mineral soils (Dettmann et al., 2014; Mualem, 1976) |
| **Compact Sand's Hydraulic Parameters** | | |
| $\theta_r$<br>$\theta_s$<br>$\alpha$<br>N | 0.0429 cm$^3$ cm$^{-3}$<br>0.365 cm$^3$ cm$^{-3}$<br>0.0535 cm$^{-1}$<br>1.666 | Optimized using inverse modeling under vegetation parameterization 3; constraints derived from measured grain size compositions via ROSETTA 2 |
| $K_s$ | 1.283 cm d$^{-1}$ | Optimized using inverse modeling; constraints from head test measurements |
| L | 0.5 | Usual value for mineral soils (Dettmann et al., 2014; Mualem, 1976) |





### 2.3 Scenario Analyses

The calibrated and validated models were then used to quantify the water budget under the historical (1972–2000) and wet and dry future climate scenarios (2072–2100). We used a constant LAI based on the temporal means of measured LAI in our
experimental setups (1.57 for non-compacted and 1.32 for compacted case models) (Fig. 2e). For Vegetation Parameterization 1, the non-compacted case model's LAI was adopted for the compacted case model (Fig. 4a, Fig. 4b). These ensure that the seasonal patterns of hydrological variables are solely brought by the climate forcings. The initial soil moisture profiles were also set at field capacity.

The future climate projection data are in the form of a 30 run ensemble from regional climate models, used to force our soil water flow models. All these runs, set at the Royal Meteorological Institute at Uccle, Brussels, Belgium (Appendix A – Fig. A1), can be considered representative for Flanders except for the coastal area (Van Schaeybroeck et al., 2021). They were generated based on the EURO-CORDEX regional climate models under the combined four Representative Concentration Pathway scenarios (RCP 2.6, 4.5, 6.0, 8.5) for the future period 2071–2100. Equal weights were given to these four RCP
scenarios. The regional climate model outputs were statistically downscaled via the quantile perturbation method of Willems & Vrac (2011) based on the historical daily meteorological observations in Uccle (01 January 1971 to 31 December 2000).

For the models' spin-up, we discarded results from January–November 1971 under each run. This is to omit undesired deep percolation recorded a few days after the start of model simulations. We then define the year 1972 as the calendar months of
December 1971 to November 1972 and so on. Thus, each run spans from the years 1972 to 2000, corresponding to the calendar months of December 1971 to November 2000. With this, we ensured equal number of data sets across seasons in our scenario analyses: 29 data sets each under winter (DJF: December-January-February), spring (MAM: March-April-May), summer (JJA: June-July-August), and autumn (SON: September-October-November). This scheme also applies for future scenarios.


To represent a wet and a dry future scenario, we used the 95 percentiles of the mean annual hydric excess $(P – ET_0)$ and deficit $(ET_0 – P)$, respectively, from the full ensemble of 30 model runs (Table 4) (Christensen and Christensen, 2003; Van Schaeybroeck et al., 2021). These 95 percentile based scenarios can be interpreted as "high-impact" scenarios, where the future is expected to lie with high likelihood between the current climate and that high-impact scenario.


**Table 4. Selected wet and dry future scenarios and specifications**

| Scenario | Main Basis for Selection | RCP | Regional Climate Model |
|---|---|---|---|
| Wet | 95 percentile mean annual hydric excess $(P – ET_0)$ | RCP4.5 | HadGEM2-CC-r1 |
| Dry | 95 percentile mean annual hydric deficit $(ET_0 – P)$ | RCP6.0 | MIROC-ESM-r1 |

The whole modeling workflow from setup to scenario analyses is summarized in Fig. 5.





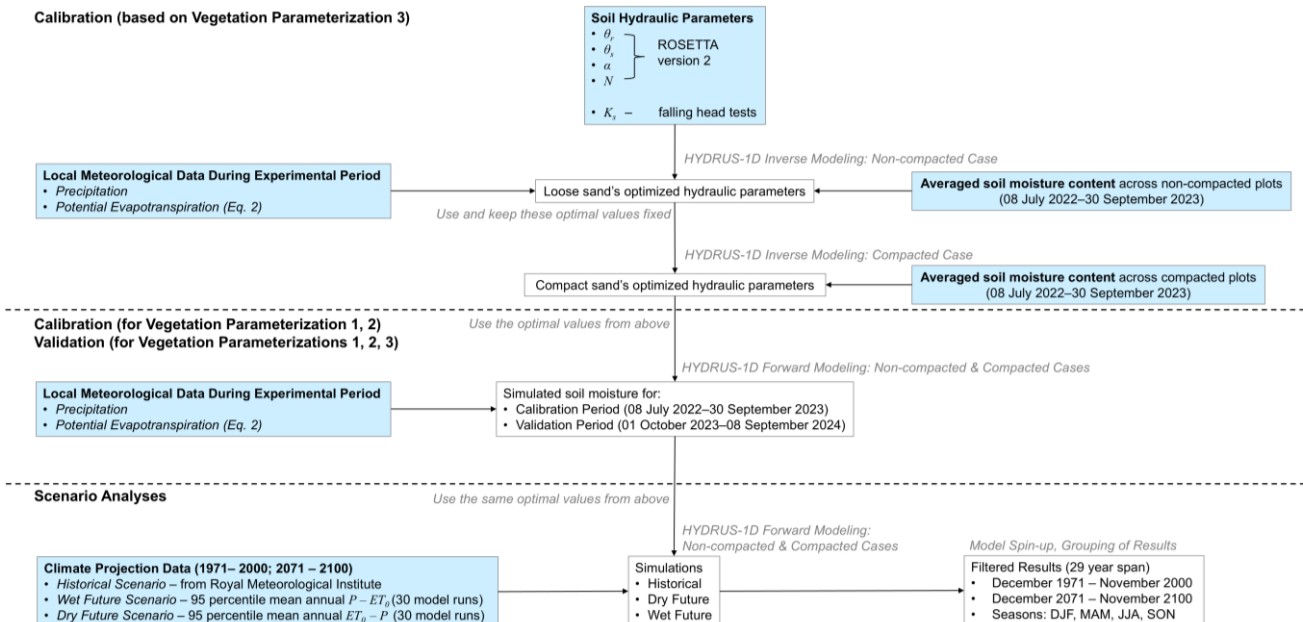


**Figure 5. The modeling workflow of this study**





# 3 Results

## 3.1 Simulated vs Observed Soil Moisture Content

Overall, the compacted setup has higher soil moisture than the non-compacted setup, as showed by both observation data and simulations (Fig. 6). At 40 cm depth, these differences are even larger. Moreover, very high soil moisture peaks are more flattened in the compacted setup, reflecting the compacted setup's frequent saturation moments (Fig. 6d). Despite the irrigation at the experiment's beginning, the non-compacted setup was drier than the compact soil. All these reflect the compact layer's role in both experimental plots and simulations.


At 10 cm depth, the models simulate the soil moisture dynamics well. However, at 40 cm depth, simulations underestimate the soil moisture (Fig. 6b, Fig. 6d). Moreover, the compacted case models simulate a faster decline in moisture content in mid-January 2023 to start of March 2023 (Fig. 6d). This might indicate that the compacted layer's estimated hydraulic conductivity from optimizations is still relatively high. Nearer to the soil surface (Fig. 6a, Fig. 6c), models involving

different root depths are closer to observations than the ones with same root depths.

At 10 cm depth (Fig. 6a), the deep rooted non-compacted case model improved the moisture content simulations in mid-May 2023 to mid-July compared to its shallow rooted counterpart. With shallow roots, the model overestimates the root water uptake, leading to its simulations' faster decline than both the observations and deeper root simulations. At 40 cm depth (Fig.

6b), the shallow rooted non-compacted case model underestimated the observed decline in moisture content in June–July 2023. Meanwhile, for the compacted case models (Fig. 6c, Fig. 6d), the differences in LAI yielded very little differences in the simulated moisture contents.



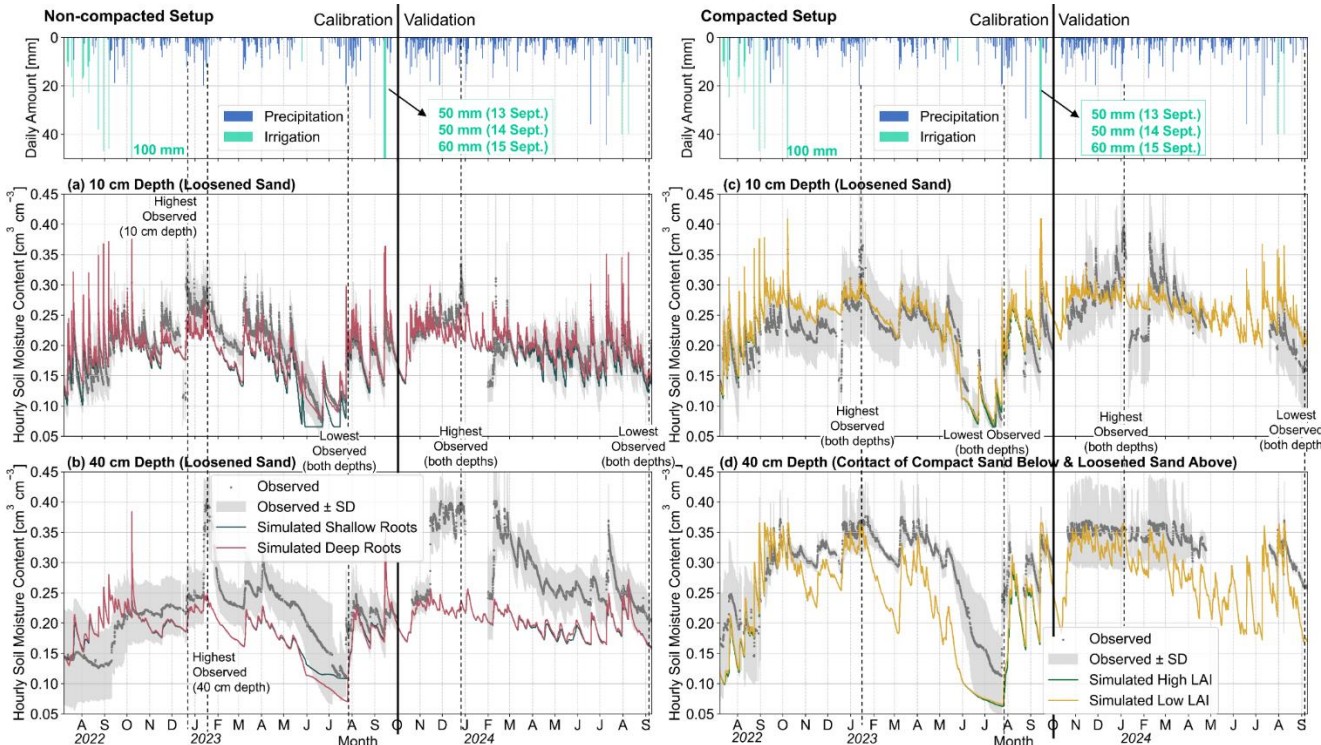

**Figure 6. Observed soil moisture content (± standard deviation (SD)) vs simulated soil moisture content time series at 10 and 40 cm depths under various vegetation parameterizations under calibration (08 July 2022–30 September 2023) and validation periods (01 October 2023–30 September 2024). Performance indicators are listed in Appendix B – Table B4.**



## 3.2 Scenario Analyses (1972–2000; 2072–2100)

Compared to the historical scenario, the wet scenario depicts slightly higher annual precipitation yet similar annual potential
evapotranspiration (Fig. 7a, Fig. 7c). Meanwhile, the dry scenario has a similar annual precipitation yet higher summer
potential evapotranspiration. Both future scenarios also show stronger precipitation seasonality (lower in summers and
higher in winters) than the historical scenario (Fig. 7b, Fig. 7d).

**Figure 7. Annual and monthly precipitation and potential evapotranspiration under the historical and wet and dry future
scenarios. Boxplots are also colored by seasons.**

**Abbreviations:** *DJF*=December-January-February*; MAM*=March-April-May; *JJA*=June-July-August; *SON*=September-October-
November



For the hydrological components (actual evapotranspiration, actual evaporation, actual transpiration, soil moisture storage above the compact layer [i.e., 0–40 cm], deep percolation), the actual fluxes are reported in Fig. 8 and Fig. 9. We also report their results in terms of the annual and monthly differences between the compacted and non-compacted case models in Fig. 10 and Fig. 11.

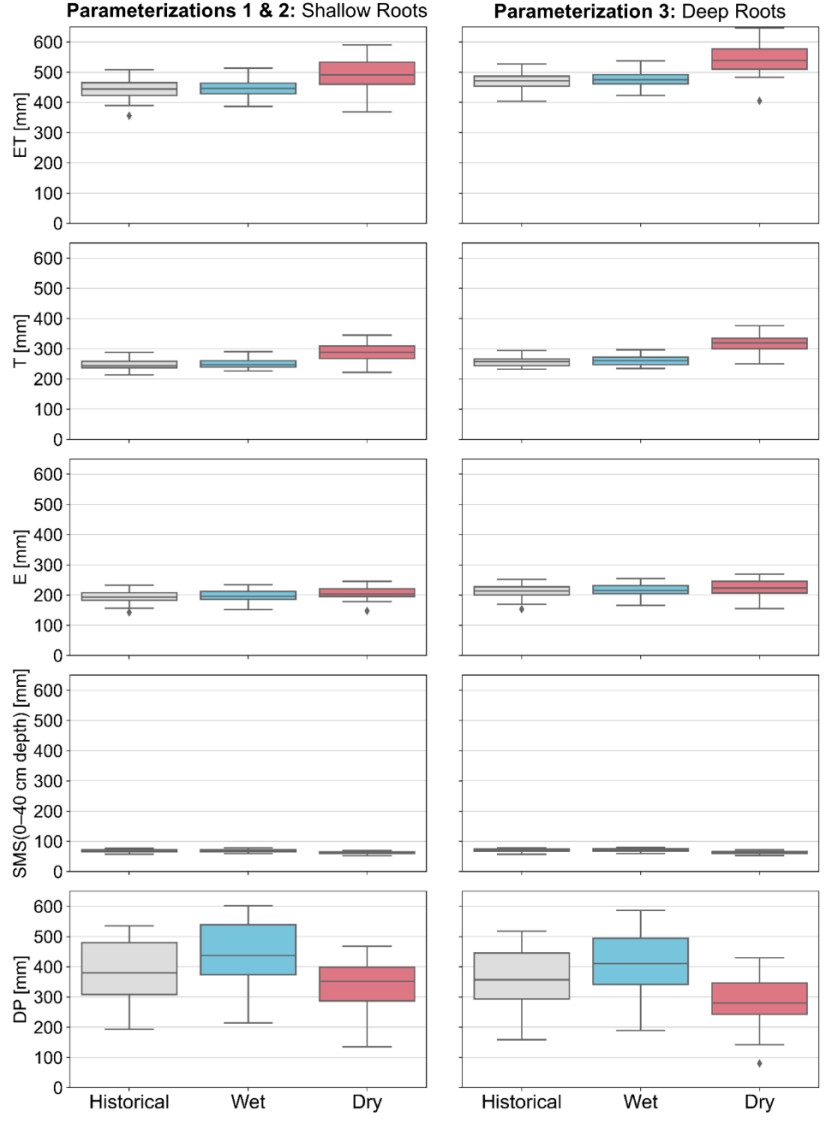

**Figure 8. Annual comparison of different hydrometeorological variables for the non-compacted case under historical and wet and dry future scenarios across the three vegetation parameterizations. All the y-axes have the same scale**

**Abbreviations: *ET*=actual evapotranspiration, *E*=actual evaporation, *T*=actual transpiration, *SMS*=mean daily soil moisture storage, *DP*=deep percolation**



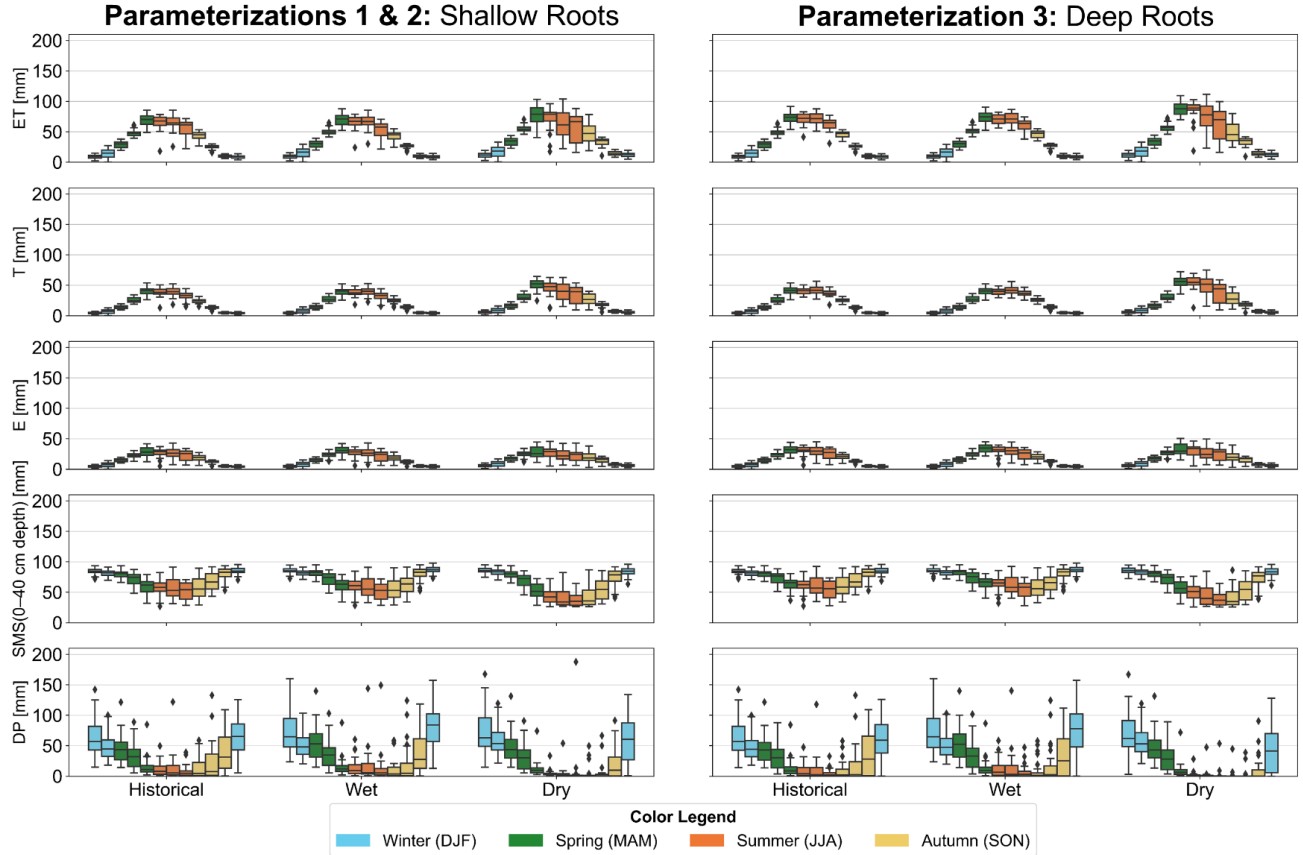

**Figure 9. Monthly comparison of different hydrometeorological variables for the non-compacted case under the historical and wet and dry future scenarios. For each scenario under a subplot, 12 boxplots are shown, representing the months of the year from January to December. Sets of boxplots are colored according to their seasons (see legend above). All the y-axes have the same scales.**

**Abbreviations: *ET*=actual evapotranspiration, *E*=actual evaporation, *T*=actual transpiration, *SMS*=mean daily soil moisture storage, *DP*=deep percolation**

With compaction, water storage in the upper soil profile increases during winter for all vegetation parameterizations and climate scenarios (Fig. 11a, Fig. 11b). However, in the dry climate scenario's summers, both the compacted and non-compacted soils' upper layers dry out to similarly low moisture contents. This is most noticeable for vegetation parameterization 3 with deeper roots and higher LAI in the non-compacted model than in the compacted soil model (Fig. 11c).





For vegetation parameterizations 1 and 2 that assumes same shallow root depths and root density profiles (Fig. 10a, Fig. 10b), actual evapotranspiration and evaporation increase, and deep percolation decreases with compaction. The lower LAI for the compacted case model in parameterization 2 leads to a lower transpiration than the non-compacted case. This is almost fully compensated by higher evaporation in the compacted case model. Thus, vegetation parameterizations 1 and 2 do not exhibit much difference on the simulated impact of compaction on deep percolation.


Under vegetation parameterization 3, the non-compacted case model's deeper rooting leads to more water uptake from the deeper subsoil. This further increases the difference in transpiration between the non-compacted and compacted case models, especially under the dry future scenario (Fig. 10c). This deeper rooting also reduced the uptake from the non-compacted case model's upper depths compared to the shallow rooting from parameterization 2 (Fig. 10b). Thus, more water remained

available for evaporation in the non-compacted soil, leading to smaller differences in evaporation losses between the compacted and non-compacted soils (Fig. 10c), than in parameterization 2 (Fig. 10b). Higher transpiration and evaporation from the non-compacted case model also led to its simulated less deep percolation than in parameterization 2. This is shown by parameterization 3's smaller difference in simulated deep percolation between the compacted and non-compacted case models (Fig. 10c). For the historical and wet climate scenarios, more deep percolation is simulated in the non-compacted

soil, but the opposite occurs for the dry climate scenario. Thus, whether deep percolation increases or decreases with compaction or after de-compaction depends on how vegetation was parameterized and can be opposite in dry or wet future climate scenarios (Fig. 10c).

In terms of seasonal dynamics, when root depth does not change with compaction (Fig. 11a, Fig. 11b), deep percolation is

reduced and delayed for the non-compacted setup after the summer season (especially in November and December). However, if root depth is also adjusted after compaction (Fig. 11c), deep percolation is reduced in the compacted setup in March and April. During this period, the compacted setup has water still draining from the subsoil while root water uptake from deeper layer in the non-compacted setup already starts, reducing deep percolation.

All these key findings of water balance changes due to compaction under various vegetation parameterizations are summarized in Fig. 12.





**Figure 10. Annual comparison of hydrological variables for the historical and wet and dry future scenarios across vegetation parameterizations (a) 1, (b) 2, and (c) 3. All the y-axes scales are ensured to be consistent. As such, outlier *ΔDP* values of -196 and -157 were excluded from parameterizations 1 and 2, respectively. *Positive values:* compacted case > non-compacted case. *Negative values:* compacted case < non-compacted case**


**Abbreviations: *ET*=actual evapotranspiration, *E*=actual evaporation, *T*=actual transpiration, *SMS*=mean daily soil moisture storage, DP=deep percolation. "*Δ*" signifies subtraction difference between compacted and non-compacted cases' calculated values.**







**Figure 11. Monthly comparison of hydrological variables for the historical and wet and dry future scenarios across vegetation parameterizations (a) 1, (b) 2, and (c) 3. For each scenario under a subplot, 12 boxplots are shown, representing the calendar months (January–December). Boxplots are also colored by seasons (see legend above). All the y-axes scales are ensured to be consistent. As such, 9 and 6 outlier *ΔDP* values < -40 were excluded from parameterizations 1 and 2, respectively, belonging to either August, October, or December. *Positive values:* compacted case > non-compacted case. *Negative values:* compacted case < non-compacted case**

**Abbreviations: *ET*=actual evapotranspiration, *E*=actual evaporation, *T*=actual transpiration, *SMS*=mean daily soil moisture storage, DP=deep percolation. *DJF*=December-January-February*; MAM*=March-April-May; *JJA*=June-July-August; *SON*=September-October-November. "*Δ*" signifies differences between calculated values of compacted and non-compacted case.**



**Figure. 12. Schematic summary of the impacts of sandy subsoil compaction to the soil hydrology's components under historical and wet scenarios according to the three vegetation parameterizations. The indicated season is when the impact is most significant in a year.**





## 4 Discussion

### 4.1 Implication on Water Resource Management

Results from our scenario analysis show the effect of both soil compaction and its interaction with the vegetation on the soil water balance (Fig. 12). As per parameterizations 1 and 2 (Fig. 10a, Fig. 10b, Fig. 11a, Fig. 11b), a trade-off seemingly exists between soil water retention for vegetation (by leaving the subsoil compacted, based on high soil moisture for the compacted case model) and maximizing groundwater recharge potential (by de-compaction, based on high deep percolation for the non-compacted case model). Considering that compaction reduces vegetation biomass production (i.e., parameterization 2) (Fig. 10b, Fig. 11b), de-compaction could promote both groundwater recharge and vegetation productivity. Finally, if the vegetation is allowed to develop a deeper root system in the de-compacted case model (i.e., parameterization 3), de-compaction will not always guarantee higher recharge (Fig. 10c). These abovementioned inferences drawn from these three vegetation parameterizations provide very different and even contrasting overviews of the agricultural water availability's dynamics and sustainability. From these could arise conflicting water resource management strategies tailored to those inferences in this climate change context.

These insights show that while sandy subsoil compaction directly affects both vegetation growth and water balance, one should not forget that the affected vegetation growth also further influences the water balance. Therefore, in hydrological studies involving (de-)compaction, vegetation growth above- and belowground should be dynamically incorporated. With this, field evidence of vegetation growth, root growth and yield, often far lacking in compaction studies, is crucial.

### 4.2 Limitations of the Study

Compared to observations at 40 cm depth, the non-compacted case's models severely underestimate the observed soil moisture during wet periods – in two weeks of January 2023 during calibration period (narrow peak) and mid-November to end of December 2023 and mid-February to March 2024 during validation period (wider peak) (Fig. 6b, Appendix A – Fig. A8). The water table could have risen temporarily high enough to saturate these loosened layers at 40 cm depth. The wider soil moisture peak during 2023–2024 is likely due to much higher precipitation (528 mm from 01 October to 31 March) than 2022–2023 (359 mm from 01 October to 31 March) in the site. Unfortunately, the 200 cm deep water table we observed from soil profiles during March 2023 and 2024 could have occurred only when the water table had already receded. In flat, low-lying areas, such shallow water table can influence soil hydrological fluxes (Groh et al., 2016) and even vegetation growth (Glanville et al., 2023; Horsnell et al., 2009; Odili et al., 2023; Ridolfi et al., 2006). With this, the feedback loop involving soil compaction, vegetation, and soil water becomes more complex, which, however, is beyond our study's scope.



We did not also indicate the runoff amounts in our main results. This is because under all scenarios, runoff only occurs for one day just on all compacted case models (i.e., 31 mm on 29 August 1996; 74 mm and 96 mm in wet and dry August 2096, respectively) whenever an extreme rainfall event occurs under very high antecedent soil moisture conditions. This means that runoff appears to be infrequently simulated, despite reported occurrences of waterlogging in even sandy subsoil
compaction (Huang and Hartemink, 2020; Polge De Combret - Champart et al., 2013). This might be due to the difficulty in simulating infiltration-excess runoff as the projected meteorological data have coarse time resolution (i.e., daily), and thus rainfall intensity is not considered (Mertens et al., 2002). This then leads to overestimated infiltration and underestimated runoff in our scenario analyses (Šimůnek and Weihermüller, 2018). Meanwhile, our hourly simulations during the experiments generated runoff for the compacted setups (Appendix B – Table B5), linked to intense irrigation events (40 mm
runoff brought by 100 mm irrigation in three hours of 07 October 2022, 27 mm runoff brought by 160 mm irrigation of which 50–60 mm is delivered for six hours every day of 13–15 September 2023 (Appendix B – Table B2)). Nevertheless, the compaction's role on runoff generation is still clear as only the compacted case models generated runoff in both the experiments and scenario analyses.

**5 Conclusion**

To assess both compaction's hydrological impact and de-compaction's effectiveness to promote infiltration and groundwater recharge, soil water flow modeling has long been performed to quantify its impacts on soil water balance. Here, for sandy subsoils, we showed that interpretations and insight from these model simulations can drastically change with the level of understanding on the vegetation's role in the study site.

By assuming same root depths and LAI, compaction promotes more soil storage and thus more evaporable water while reducing and delaying deep percolation. Lower LAIs for the compacted setups further lead to higher evaporation yet lower transpiration than the compacted case with unadjusted LAIs. Finally, if root depths increased also for non-compacted setups, evaporation still increased for the compacted case while transpiration decreased. However, deep percolation can increase or decrease with compaction depending on the year (Fig. 12). Thus, having different results across various vegetation
parameterizations highlights the need for more accurate parameterization of vegetation's growth to achieve more robust and confident conclusions. In other words, to properly assess sandy subsoil (de-)compaction's hydrological impacts, the vegetation's role must be clearly understood.

Climate change would likely negatively impact crop production as it could promote more and longer dry (droughts) (Cotrina
Cabello et al., 2023; Li et al., 2009; Shanker et al., 2014) and even very wet periods (waterlogging) (Fischer et al., 2023; Tian et al., 2021; Yang et al., 2024). These crop's long-term responses under these contrasting extreme events could also be



considered in the hydrological model simulations of future scenarios. This could refine our current understanding of future agricultural water balance even further, including evaporation losses and groundwater recharge.



**Appendix A: Additional Figures**

**Figure A1. Location map showing the study area, the location region of soil map (Appendix A – Fig. A4), and nearby countries (top inset) and municipality (bottom inset). Coordinate system: EPSG 31370 - BD72/Belgian Lambert 72. Basemap: Bing Virtual Earth (©Microsoft, 2012)**





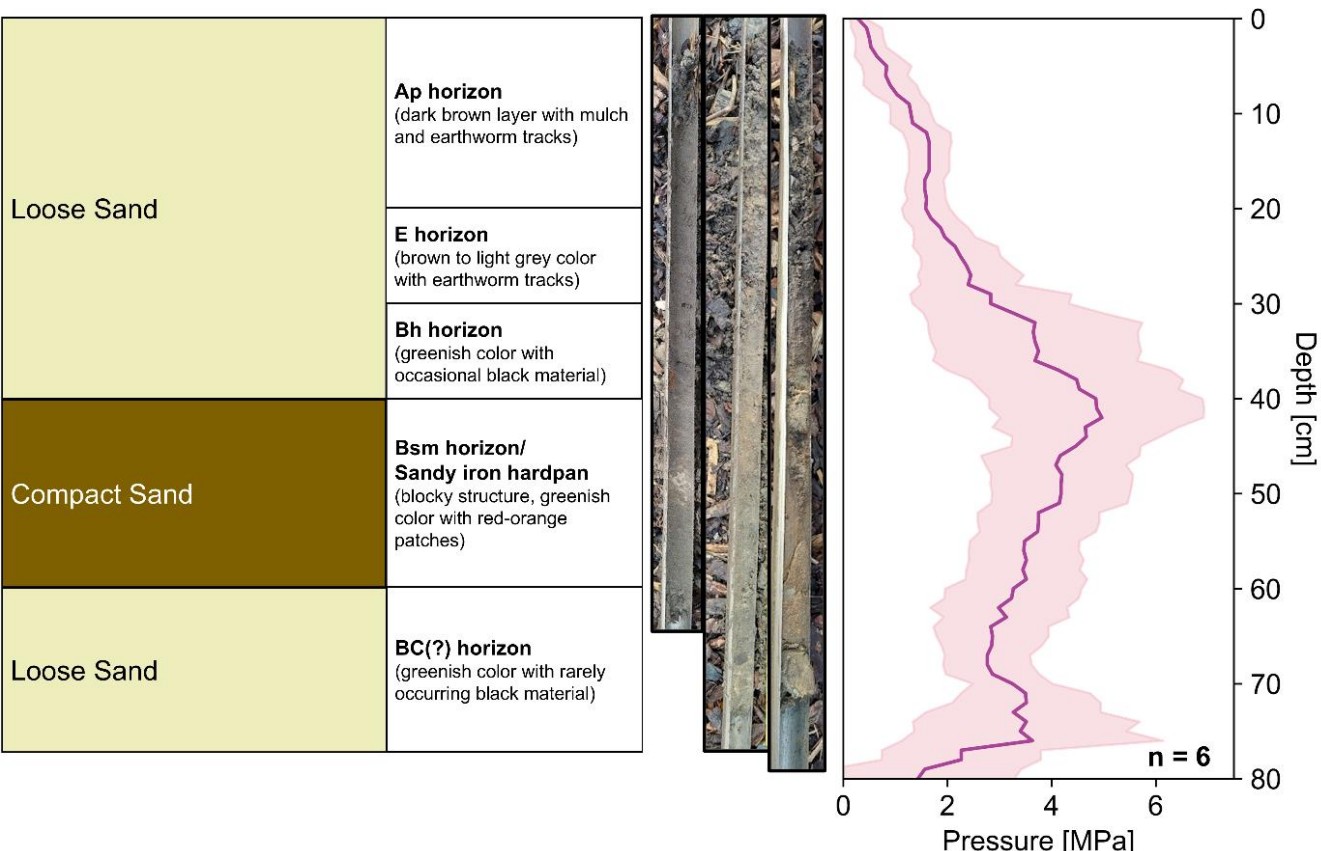


**Figure A2.** Characterized soil profiles taken outside the plots and the corresponding penetrologger profile





Figure A3. Soil types present in the vicinity (based on the boxed region in Appendix A – Fig. A1). The soil map data are based on Dondeyne et al. (2015). Coordinate system: *EPSG 31370 - BD72/Belgian Lambert 72.*

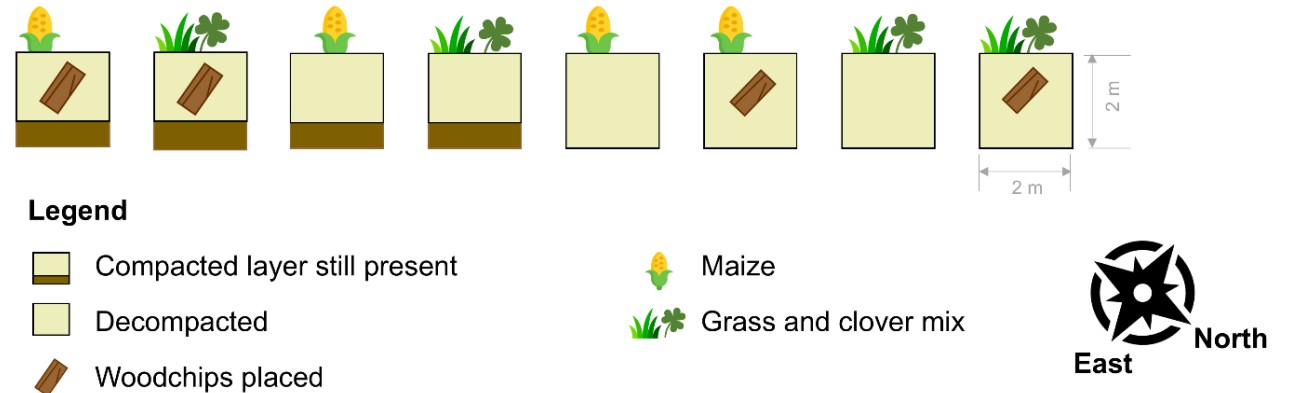

Figure A4. Experimental Plots in the Site, showing various treatments such as presence and absence of the compact layer, vegetation type (maize, grass-clover mixture), and application of woodchips for certain plots.



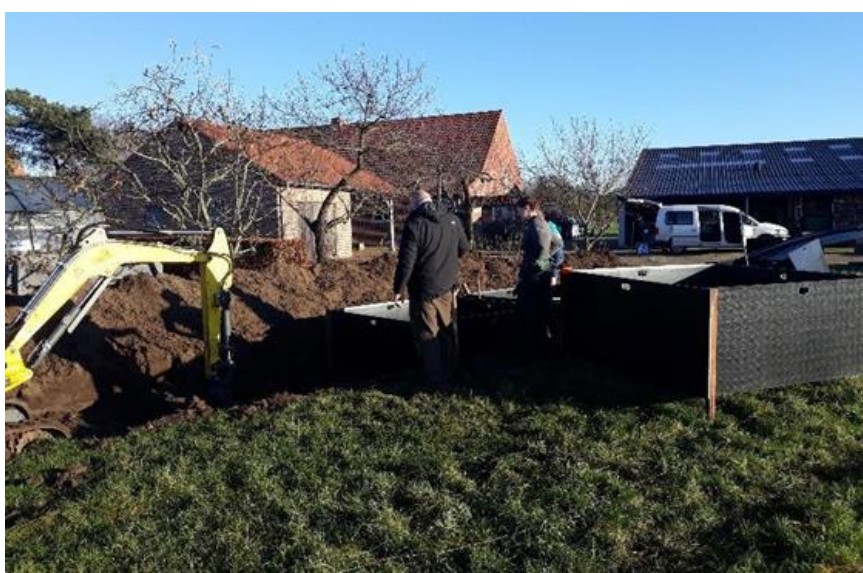


**Figure A5. Excavation works during set up of experimental site, showing the black impermeable plastic boxes and the crane used for de-compaction**





**Figure A6. General Timeline of the Experimental Activities**





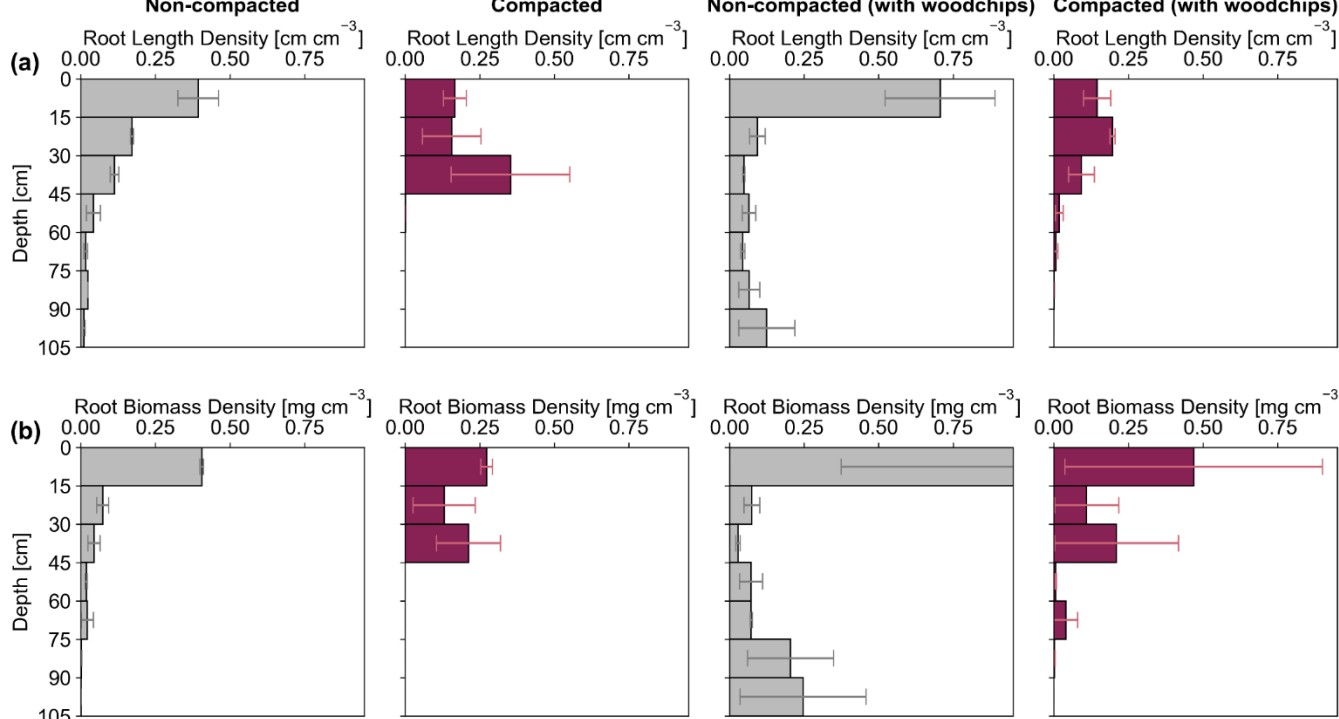

**Figure A7. (a) Mean root length and (b) biomass densities (± standard deviation) across individual plots of grass-clover mixtures (n = 2). Note that the non-compacted plot with woodchips' 0 –15 cm depth has a mean root biomass density exceeding 1.00 (i.e., 2.83 ± 2.46)**



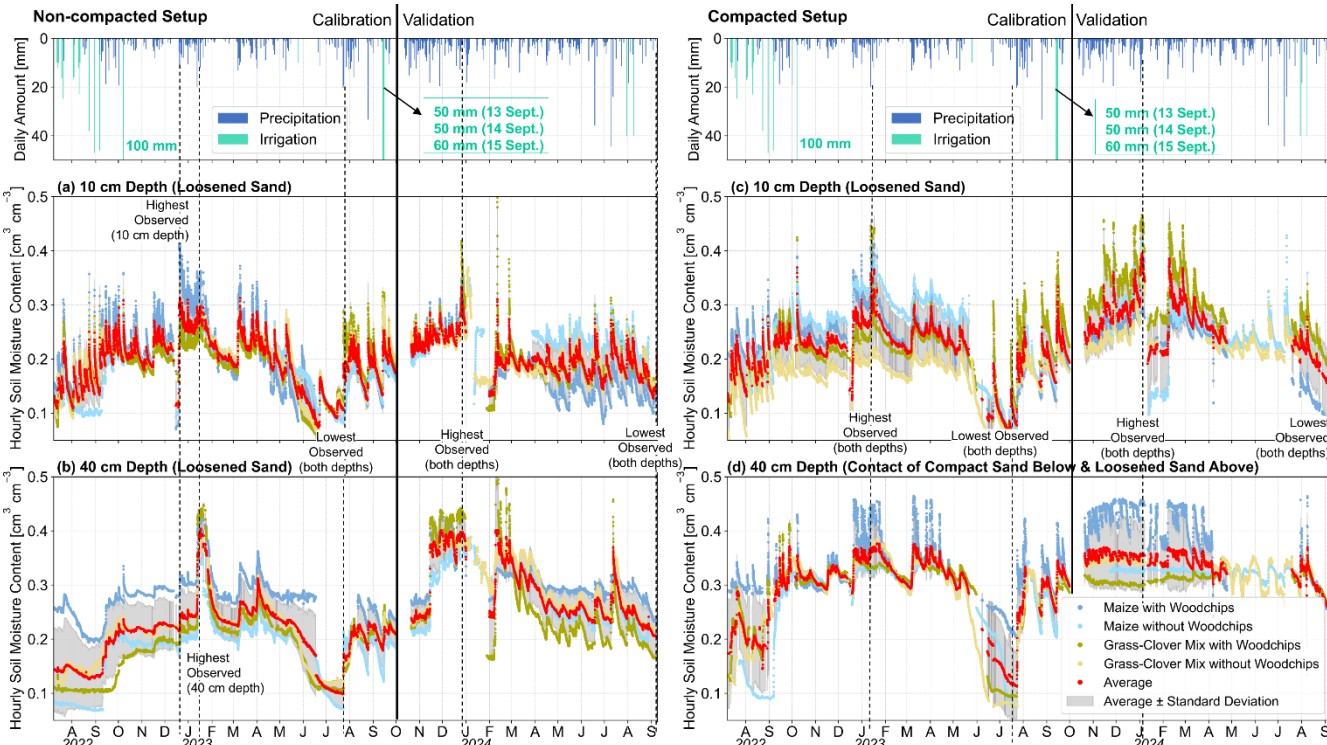


**Figure A8. Soil moisture content measurements for each plot and averages across them.**



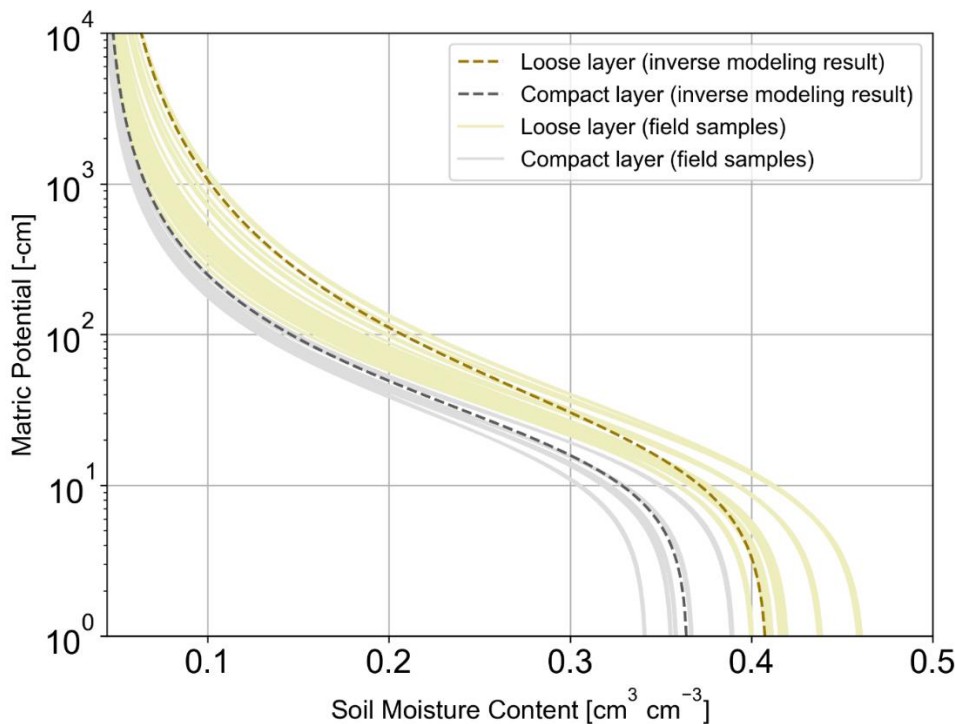

**Figure A9. Water retention curves for loosened and compact sand layers based on optimization from inverse modeling and on field samples whose properties were inferred from ROSETTA.**

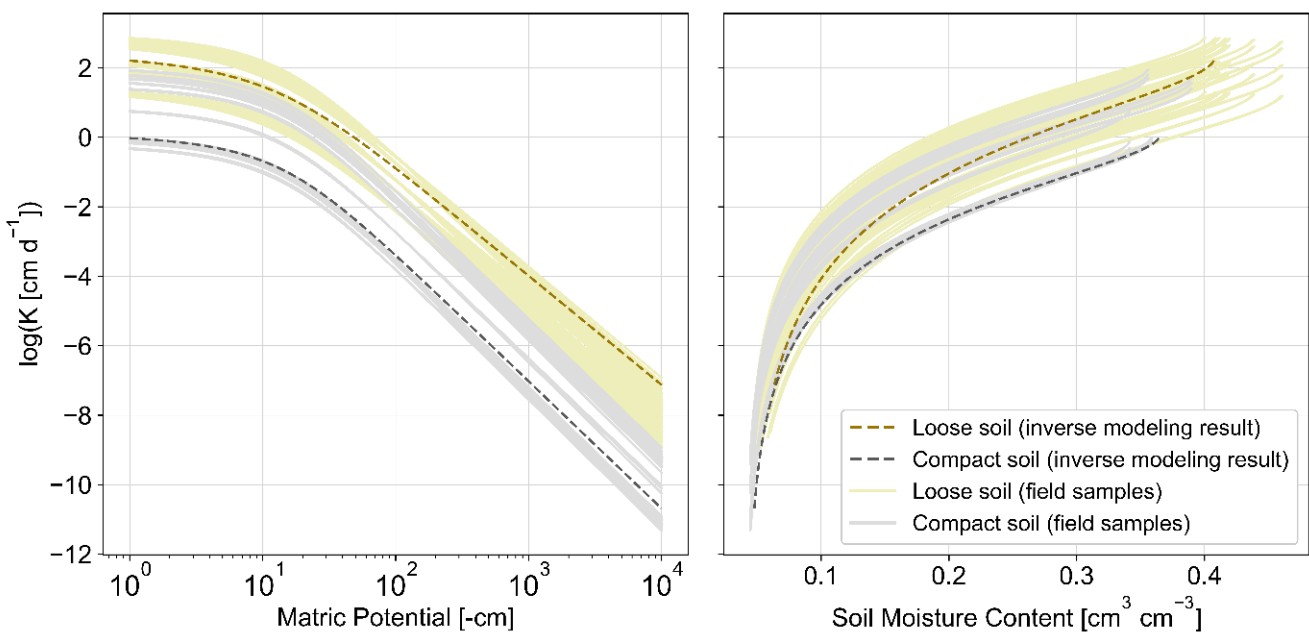


**Figure A10. Conductivity curves (log K vs pF (left), log K vs θ (right)) for both loosened and compact sand layer based on optimization from inverse modeling and on field samples whose properties were inferred from ROSETTA and $K_s$ measurements (falling head method)**



## Appendix B: Additional Tables


**Table B1. Irrigation amounts applied during the study period (July 2022–September 2024) and based historical events in Herentals (closest active meteorological station from our study site) (MOW-HIC et al., 2021)**

| Date and Time of Irrigation | Irrigation amount [mm] | Dates of Actual Historical Summer Rainfall Event in Herentals that Produced these Amounts |
|---|---|---|
| 12 July 2022, 12:00–13:00 | 10 | |
| 13 July 2022, 08:00–09:00 | 12 | 26–28 July 2008 |
| 14 July 2022, 11:00–12:00 | 10 | |
| 18 July 2022, 10:00–11:00 | 16.5 | 12-14 July 2005 |
| 20 July 2022, 10:00–12:00 | 24 | |
| 02 August 2022, 12:00–13:00 | 17 | 23–25 June 2016 |
| 03 August 2022, 07:00–08:00 | 16 | |
| 09 August 2022, 12:00–13:00 | 12 | |
| 10 August 2022, 08:00–10:00 | 23 | 11–13 June 2016 |
| 11 August 2022, 09:00–10:00 | 10 | |
| 23 August 2022, 10:00–13:00 | 38 | 04 June 2021 |
| 30 August 2022, 08:00–11:00 | 47 | 18 August 2011 |
| 06 September 2022, 08:00–11:00 | 46 | 15 July 2021 (time of 2021 summer European flood) |

**Table B2. Other applied irrigation amounts applied (not based on historical events)**

| Date and Time of Irrigation | Irrigation amount [mm] | Reason for Applying |
|---|---|---|
| 07 October 2022, 14:00–18:00 | 100 | Simulate the start of long-term wet period with very high soil moisture content |
| 25 May 2023, 10:00–11:00 | 10 | At that time, 35 maize seeds were sown in each maize plot and thus had to be irrigated to guarantee growth. Thus, the same amount was irrigated to grass-clover plots to ensure consistent treatment. |
| 13 September 2023, 13:00–18:00 | 50 | Generated a high yet unrecorded 3 day rainfall event to simulate future wet summer scenarios |
| 14 September 2023, 10:00–15:00 | 50 | |
| 15 September 2023, 11:00–18:00 | 60 | |
| 30 July 2024, 10:00–14:00 | 40 | To alleviate potential drought stress for the vegetation |
| 08 August 2024, 10:00–14:00 | 40 | |


**Table B3. Constraints used for inverse modeling**

| Loosened Sand | $\theta_r$ [cm³ cm⁻³] | $\theta_s$ [cm³ cm⁻³] | $\alpha$ [cm⁻¹] | $n$ [-] | $K_s$ [cm d⁻¹] |
|---|---|---|---|---|---|
| Minimum | 0.0412 | 0.401 | 0.0544 | 1.457 | 26.7 |
| Maximum | 0.0427 | 0.463 | 0.0616 | 1.675 | 977.8 |
| Optimal | 0.0427 | 0.409 | 0.0544 | 1.457 | 297.672 |

| Compact Sand | $\theta_r$ [cm³ cm⁻³] | $\theta_s$ [cm³ cm⁻³] | $\alpha$ [cm⁻¹] | $n$ [-] | $K_s$ [cm d⁻¹] |
|---|---|---|---|---|---|
| Minimum | 0.0401 | 0.342 | 0.0535 | 1.666 | 0.634 |
| Maximum | 0.0429 | 0.391 | 0.0574 | 1.738 | 111 |
| Optimal | 0.0429 | 0.365 | 0.0535 | 1.666 | 1.283 |





**Table B4. Performance indicators of the models based on the presence of compaction, observation point depths, and model parameterizations**

**Calibration Period (08 July 2022 – 30 September 2023)**

| Case | Non-compacted | | | | Compacted | | | |
|---|---|---|---|---|---|---|---|---|
| **Observation Depth** | **10 cm** | | **40 cm** | | **10 cm** | | **40 cm** | |
| **Model Parameterization** | **Shallow Roots** | **Deep Roots** | **Shallow Roots** | **Deep Roots** | **High LAI** | **Low LAI** | **High LAI** | **Low LAI** |
| Mean SMC, observed | 0.191 | 0.191 | 0.208 | 0.208 | 0.211 | 0.211 | 0.282 | 0.282 |
| Mean SMC, simulated | 0.176 | 0.185 | 0.186 | 0.185 | 0.229 | 0.231 | 0.231 | 0.233 |
| $R^2$ | 0.687 | 0.661 | 0.212 | 0.224 | 0.831 | 0.821 | 0.719 | 0.719 |
| NSE | 0.560 | 0.644 | -0.052 | -0.112 | 0.672 | 0.657 | -0.024 | 0.027 |
| RMSE | 0.032 | 0.029 | 0.050 | 0.052 | 0.030 | 0.031 | 0.069 | 0.067 |
| MAE | 0.027 | 0.023 | 0.041 | 0.043 | 0.024 | 0.025 | 0.059 | 0.057 |
| ME | 0.014 | 0.005 | 0.021 | 0.023 | -0.018 | -0.020 | 0.051 | 0.049 |

**Validation Period (01 October 2023 – 08 September 2024)**

| Case | Non-compacted | | | | Compacted | | | |
|---|---|---|---|---|---|---|---|---|
| **Observation Depth** | **10 cm** | | **40 cm** | | **10 cm** | | **40 cm** | |
| **Model Parameterization** | **Shallow Roots** | **Deep Roots** | **Shallow Roots** | **Deep Roots** | **High LAI** | **Low LAI** | **High LAI** | **Low LAI** |
| Mean SMC, observed | 0.199 | 0.199 | 0.286 | 0.286 | 0.260 | 0.260 | 0.338 | 0.338 |
| Mean SMC, simulated | 0.195 | 0.202 | 0.201 | 0.203 | 0.267 | 0.267 | 0.299 | 0.299 |
| $R^2$ | 0.566 | 0.562 | 0.371 | 0.364 | 0.507 | 0.508 | 0.803 | 0.803 |
| NSE | 0.499 | 0.540 | -1.998 | -1.901 | 0.418 | 0.418 | -2.486 | -2.482 |
| RMSE | 0.023 | 0.022 | 0.096 | 0.094 | 0.036 | 0.036 | 0.046 | 0.046 |
| MAE | 0.018 | 0.015 | 0.085 | 0.083 | 0.030 | 0.030 | 0.040 | 0.040 |
| ME | 0.004 | -0.003 | 0.085 | 0.083 | -0.008 | -0.008 | 0.039 | 0.039 |

**Abbreviations: SMC=Soil Moisture Content [$cm^3\ cm^{-3}$]; $R^2$=Coefficient of Determination; NSE=Nash-Sutcliffe Efficiency; RMSE=Root Mean Squared Error [$cm^3\ cm^{-3}$]; MAE=Mean Absolute Error [$cm^3\ cm^{-3}$]; ME=Mean Error [$cm^3\ cm^{-3}$]**



**Table B5. Water budget results for the three vegetation parameterizations during the experimental period. All units are in mm.**

**Calibration Period (08 July 2022 – 30 September 2023)**

| Vegetation Parameterization | Case Model | Inflows | | Outflows | | | | | Storages | | |
|---|---|---|---|---|---|---|---|---|---|---|---|
| | | P | Irrigation | E | T | ET | DP | R | SMS_0 | SMS_F | Net |
| (1) Same High LAI, Same Shallow Roots | Non-compacted | 837 | 552 | 357 | 360 | 717 | 623 | 0 | 149 | 198 | 0 |
| | Compacted | 837 | 552 | 397 | 367 | 764 | 482 | 66 | 165 | 242 | 0 |
| (2) Lower LAI (Compacted), Same Shallow Roots | Non-compacted | 837 | 552 | 357 | 360 | 717 | 623 | 0 | 149 | 198 | 0 |
| | Compacted | 837 | 552 | 452 | 307 | 759 | 486 | 67 | 165 | 242 | 0 |
| (3) Lower LAI (Compacted), Deeper Roots (Non-compacted) | Non-compacted | 837 | 552 | 380 | 369 | 749 | 591 | 0 | 149 | 198 | 0 |
| | Compacted | 837 | 552 | 452 | 307 | 759 | 486 | 67 | 165 | 242 | 0 |

**Validation Period (01 October 2023 – 08 September 2024)**

| Vegetation Parameterization | Case Model | Inflows | | Outflows | | | | | Storages | | |
|---|---|---|---|---|---|---|---|---|---|---|---|
| | | P | Irrigation | E | T | ET | DP | R | SMS_0 | SMS_F | Net |
| (1) Same High LAI, Same Shallow Roots | Non-compacted | 923 | 80 | 229 | 287 | 516 | 522 | 0 | 198 | 163 | 0 |
| | Compacted | 923 | 80 | 255 | 287 | 542 | 506 | 0 | 242 | 197 | 0 |
| (2) Lower LAI (Compacted), Same Shallow Roots | Non-compacted | 923 | 80 | 229 | 287 | 516 | 522 | 0 | 198 | 163 | 0 |
| | Compacted | 923 | 80 | 262 | 280 | 542 | 506 | 0 | 242 | 197 | 0 |
| (3) Lower LAI (Compacted), Deeper Roots (Non-compacted) | Non-compacted | 923 | 80 | 239 | 287 | 526 | 512 | 0 | 198 | 163 | 0 |
| | Compacted | 923 | 80 | 262 | 280 | 542 | 506 | 0 | 242 | 197 | 0 |

**Abbreviations:** *P*=precipitation, *E*=actual evaporation, *T*=actual transpiration, *ET*=actual evapotranspiration, *DP*=deep percolation, *R*=runoff, $SMS_0$=initial soil moisture storage, $SMS_0$=initial soil moisture storage=final soil moisture storage, *Net*=(P + 
*Irrig) − (ET + DP + R) − (SMS_F − SMS_0)*

**Code Availability**

We used the software HYDRUS-1D, version 4.17.0140, publicly available for download in the PC-PROGRESS Site
(https://www.pc-progress.com/en/Default.aspx?H1d-downloads) (Šimůnek et al., 2008, 2012, 2016). This includes the
inverse solution module, which allows for inverse modeling.

**Data Availability**

Daily meteorological observation data (interpolated in a 25 km square grid from 1979 to 2023), used to describe the general
climate context of our study site (Fig. 2a, Fig. 2b), are publicly available at Agri4Cast Data
(https://agri4cast.jrc.ec.europa.eu/DataPortal/RequestDataResource.aspx?idResource=7&o=d) with the grid number 103095
(Toreti, 2014).  The weather data recorded from Herentals, Belgium (5.5 km away from the site), used to fill in the missing
data of our site's local weather data, are also available in the waterinfo site (https://waterinfo.vlaanderen.be/Meetreeksen)
(MOW-HIC et al., 2021). The amount of irrigated water we applied in the experiments, along with the input values for the
inverse modeling, models' performance indicators, and water balance results during the experiments, are all in Appendix B.
The rest of our field, simulation, and climate projection raw data will be made available after publishing.

**Author Contributions**

Conceptualization: JS; Data Curation: JGP; Formal Analysis: JGP, OADS, JS, JV, SG, PW; Investigation: JGP, OADS, RD,
JS; Methodology: JS, JGP, JV, SG, PW; Visualization: JGP, JS, JV, SG; Supervision: JGP, JS, JV, SG; Writing – original
draft preparation: JGP; Writing – review: all authors.

**Competing Interests**

At least one of the co-authors is a member of the editorial board of SOIL.

**Disclaimer**

Publisher's note: Copernicus Publications remains neutral with regard to jurisdictional claims made in the text, published
maps, institutional affiliations, or any other geographical representation in this paper. While Copernicus Publications makes
every effort to include appropriate place names, the final responsibility lies with the authors.



**Acknowledgements**

We would like to express our appreciation to farmers Karel and Gert Willems for providing the location for the experimental setup and their generous hospitality. We would also like to acknowledge the Flemish Region's Blue Deal Program and the province of Antwerp for funding the materials and infrastructure works for the experimental setup. We would also like to thank the ECOSPHERE Research Group of the Department of Biology of University of Antwerp for assisting in the installation and in collecting field data and performing experiments. Some figures in this manuscript are made using Python scripts, edited using ChatGPT.

**Financial support**

Funding of researchers and monitoring equipment: Strategic Basic Research project TURQUOISE: BLUE-GREEN STRATEGIES FOR CLIMATE CHANGE ADAPTATION, funded by Research Foundation – Flanders (FWO) (Grant Number S008122N), From 01 October 2021 to 30 September 2025





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

910     .