# Peer review of "Quantifying hydrological impacts of compacted sandy subsoils using soil water flow simulations: the importance of vegetation parameterization"

_EGUsphere, 2025_

## Author Response (AR1)

**Quantifying hydrological impacts of compacted sandy subsoils using soil water flow simulations: the importance of vegetation parameterization**

Jayson Gabriel Pinza[1,2], Ona-Abeni Devos Stoffels[1], Robrecht Debbaut[1], Jan Staes[1], Jan Vanderborght[2,3], Patrick Willems[4], Sarah Garré[5]

[1]ECOSPHERE Research Group, Department of Biology, University of Antwerp, 2610 Antwerp, Belgium
[2]Division of Soil and Water Management, Department of Earth and Environmental Sciences, KU Leuven, B-3001 Leuven-Heverlee, Belgium
[3]Agrosphere Institute, IBG-3, Forschungszentrum Jülich GmbH, 52428 Jülich, Germany
[4]Urban and River Hydrology and Hydraulics Section, Department of Civil Engineering, KU Leuven, B-3001 Leuven-Heverlee, Belgium
[5]Flanders Research Institute for Agricultural, Fisheries and Food Research (ILVO), 9090 Melle, Belgium

*Correspondence to*: Jayson Gabriel Pinza (jaysonpinza@gmail.com)

**Point-by-point Response to the Reviews**

**Reviewer 1**

We thank the Reviewer for dedicating time and effort to review our manuscript. The comments helped make our paper more concise. We are also grateful for the kind words in our work.

Please see below for our responses on each comment

**Reviewer 1, Comment 1:**

*/- Figures 1 and 2 and Tables 1 and 2 are not essential and could be omitted. They seem more suitable for a review paper than for the current study.*

**Our Response:**

We omitted Figure 1 and Table 1 in the main text and transferred them to the Appendix Section. Reviewer 2 also similarly suggested to omit Figure 1 and Table 1.

For Figure 2, which involves the graphs of boxplots and time series of meteorological variables and LAI, we omitted everything EXCEPT FOR Figures 2c (time series of P, ET, irrigation, temperature) and 2e (LAI time series). We deemed 2c and 2e to be significant in the paper because 2c describes the weather conditions during the experiment period while 2e visualizes the vegetation dynamics throughout the experiment. Moreover, the data from Figures 2c and 2e also served as inputs to our calibrated and validated HYDRUS models.

With these suggestions, Figures 2c and 2e now become the new Figure 1a and 1b.

For the whole Table 2, we also believe it is significant, and we prefer to retain it. As we want our experiment to be as reproducible as possible for other researchers, showing these soil hydraulic properties helps them know the characteristics of our experimental setups. Moreover, these measurements also indicate the presence of subsoil compaction (i.e., higher bulk density, lower saturated hydraulic conductivity).

**Reviewer 1, Comment 2:**

*/- Please clarify the irrigation method used to apply the water amounts, and explain how these amounts were estimated.*

**Our Response:**

We irrigated by hand using regular 10-liter graduated watering cans with sprinkler heads. We used these graduations (1 graduation = 1 liter) as guide to estimate the irrigation amounts. By simply dividing the total volume in the watering can/s by the plot area (2 meter x 2 meter), we obtain the irrigation amounts in millimeters.

In line with this comment, we added the text below in line 165-180:

"Irrigation is performed by sprinkling using 10 liter graduated watering cans."

Reviewer 2: Fera Cleophas

We thank the Reviewer for spending time and effort reviewing our manuscript. The comments helped us smoothen the flow of discussion in our paper and ensure a more formal and objective tone throughout. We also appreciate the kind words in our work.

Please see below for our responses on each comment.

Reviewer 2, Comment 1:

*The authors may consider removing Figure 1 and Table 1 from the main body of the paper, summarizing their key points in a short paragraph within the introduction or moving them to an appendix or supplementary material if deemed necessary for completeness.*

Our Response:

We omitted Figure 1 and Table 1 in the main text and transferred them to the Appendix. Reviewer 1 also suggested omitting them.

We believe, however, that their key points were already summarized in lines 50-95 (2nd to 4th paragraphs of the introduction). Thus, we did not add a separate short summative paragraph anymore.

Reviewer 2, Comment 2:

*The manuscript frequently uses first-person pronouns such as "we" and "our" (e.g., lines 87, 88, 89, 145, 171, 175, 285, 293, 521, 528). While this is acceptable in some journals, I recommend replacing these with more objective constructions (e.g., "this study", "the present work", or "the model was calibrated") to maintain a more formal and academic tone, especially if this aligns with the journal's style guidelines.*

Our Response:

We revised all these lines accordingly by replacing the 1st-person pronouns with more objective constructions as suggested.

Reviewer 2, Comment 3:

*line 504 -505: The phrase "one should not forget" in this sentence introduces a conversational tone that may not align with the formal style expected in scientific writing. It would be more appropriate to rephrase this part of the sentence to maintain objectivity and consistency with the overall academic tone of the manuscript.*

Our Response:

We simply omitted the clause "one should not forget that" to sound more objective while retaining the idea. Thus, the new sentence becomes (see line 524-530 of marked up document):

"These insights show that while sandy subsoil compaction directly affects both vegetation growth and water balance, the affected vegetation growth also further influences the water balance."

Reviewer 2, Comment 4:

*Line 550- Please remove the citations from the Conclusion section. This part should focus on synthesizing the study's findings and implications without introducing or referencing external sources. If these citations are important to retain, consider moving them to the Discussion section where prior studies are typically discussed in relation to the current work.*

Our Response:

We transferred the whole paragraph from Conclusion section to the Discussion as last paragraph of the subsection "Implications on Water Resource Management". With this, we can retain the citations.